# Optimization Guarantees for Square-Root Natural-Gradient Variational Inference

**Navish Kumar**                                                          *navish.kumar@unibas.ch*
*University of Basel, Basel, Switzerland*
*Department of Mathematics and Computer Science*

**Thomas Möllenhoff**                                          *thomas.moellenhoff@riken.jp*
*RIKEN Center for AI Project, Tokyo, Japan*

**Mohammad Emtiyaz Khan**                                          *emtiyaz.khan@riken.jp*
*RIKEN Center for AI Project, Tokyo, Japan*

**Aurelien Lucchi**                                                    *aurelien.lucchi@unibas.ch*
*University of Basel, Basel, Switzerland*
*Department of Mathematics and Computer Science*

**Reviewed on OpenReview:** *https://openreview.net/forum?id=OMOFmb6ve7*

## Abstract

Variational inference with natural-gradient descent often shows fast convergence in practice, but its theoretical convergence guarantees have been challenging to establish. This is true even for the simplest cases that involve concave log-likelihoods and use a Gaussian approximation. We show that the challenge can be circumvented for such cases using a square-root parameterization for the Gaussian covariance. This approach establishes novel convergence guarantees for natural-gradient variational-Gaussian inference and its continuous-time gradient flow. Our experiments demonstrate the effectiveness of natural gradient methods and highlight their advantages over algorithms that use Euclidean or Wasserstein geometries.

## 1 Introduction

Variational inference (VI) is widely used in many areas of machine learning and can provide a fast approximation to the posterior distribution (Jaakkola & Jordan, 1997; Wainwright et al., 2008; Graves, 2011; Kingma & Welling, 2013). VI works by formulating Bayesian inference as an optimization problem over a restricted class of distributions which is then solved, for example, by using gradient descent (GD) (Graves, 2011; Ranganath et al., 2014; Blundell et al., 2015). A faster alternative is to use natural-gradient descent (NGD) (Amari, 1998) which exploits the information geometry of the posterior approximation and often converges much faster than GD, for example, see Figure 1(a), Khan & Nielsen (2018, Fig. 1b), or Lin et al. (2019, Fig. 3). The updates can often be implemented using message passing (Knowles & Minka, 2011; Hoffman et al., 2013; Khan & Lin, 2017) and sometimes even as Newton's method or deep-learning optimizers (Khan et al., 2017; 2018; Salimans & Knowles, 2013). Due to this, there has been a lot of interest in NGD based VI (Honkela et al., 2008; Hensman et al., 2012; Salimbeni et al., 2018; Osawa et al., 2019; Adam et al., 2021; Khan & Rue, 2023).

Despite their fast convergence, little has been done to understand the theoretical convergence properties of NGD, and its continuous counterpart, which we call natural-gradient (NG) flow. Instead, most works focus on GD (Alquier et al., 2016; Domke, 2019; 2020; Domke et al., 2024; Kim et al., 2024). For NGD, Khan et al. (2016) derived convergence results for general stochastic, non-convex settings. Still, their convergence rates are slow and there is plenty of room for improvement, especially for concave log-likelihoods. For such

likelihoods, recently, Chen et al. (2023) studied the continuous-time flow. There are also some studies on stochastic NG variational inference; for example, Wu & Gardner (2024) prove non-asymptotic convergence rates for conjugate likelihoods. However, the conjugacy assumption limits their applicability to more general settings. Both in discrete and continuous time, however, there is a gap between theory and practice for convergence guarantees of NGD.

A major difficulty in obtaining stronger convergence guarantees for NGD and NG flow lies in a subtle technical issue: commonly used parameterizations can often destroy the concavity property of the log-likelihood (Chérief-Abdellatif et al., 2019, Sec. 4.3). It is well-known that the expectation of a concave log-likelihood for a Gaussian approximation is *not* concave with respect to either the covariance or precision matrix, but only with respect to the Cholesky factor of the covariance (Challis & Barber, 2013). Therefore, commonly used parameterizations, such as natural or expectation parameterization of the exponential family, destroy the concavity property and prohibit the application of tools from convex optimization to analyze the properties of NGD. Our goal in this paper is to address this issue.

We circumvent this problem by considering variational-Gaussian inference that is defined by the square root of the Gaussian covariance matrix. We show that discretizing the NG flow with this square-root parameterization results in an NGD update based on the Cholesky factor. This idea was proposed by Tan (2025), though it was also previously explored in a broader context by Lin et al. (2023; 2024). Our method extends this idea to include any square-root parameterization. This way of parameterizing helps maintain the concavity of log-likelihood functions, which in turn facilitates convergence guarantees for both NG flow and NGD. The key contributions of our work are outlined as follows:

- We prove that the NG flow exhibits an exponential convergence rate for a strongly concave log-likelihood, as stated in Theorem 1. When discretized using the square-root parameterization, this aligns with the update provided by Tan (2025, Theorem 1). The proof of convergence hinges on establishing a Riemannian Polyak-Łojasiewicz (PL) inequality, which is shown to be valid when the lowest eigenvalue of the metric tensor is bounded.

- We prove an exponential rate of convergence (Theorem 2) for NGD using Cholesky and under strongly-concave log-likelihood; the proof extends to any square-root parameterization.

- We present empirical results showcasing the fast convergence of NGD, attributed to its Newton-like update. These results are illustrated in Figure 2 and Figure 3. It is important to highlight that our experiments employ the piecewise bounds described in (Marlin et al., 2011) for computing expectations. This raises the question of how the resulting bias affects NGD convergence. To address this, we include a discrete-time analysis with bias in our supplementary section, see Theorem 3 in Appendix D. The reason for our preference for the piecewise over the stochastic implementation is justified in Appendix E.

Our proof addresses the additional challenges posed by the non-smoothness of entropy and does not require any proximal or projection operators such as those used by Domke (2019); Domke et al. (2024). Our results can be useful for further research on proving such results for generic NGD updates. The use of square-roots could lead to suboptimal convergence, for instance, it may not converge in one step for quadratic problems (as shown in Figure 1(a)), but this only implies that even better results are possible for the general case. We provide an insight into why this happens in Section 3.5, noting that square-root-based updates essentially compute an approximate estimate of the inverse of the preconditioning matrix that is used in Newton's method.

## 2 Variational Inference

The goal of variational inference is to approximate the posterior distribution by a simpler distribution, such as the Gaussian distribution. Given a Bayesian model with $n$ likelihoods, denoted by $p(\mathcal{D}_i|\boldsymbol{\theta})$ for the $i$'th data example with parameter $\boldsymbol{\theta}$, and a prior $p(\boldsymbol{\theta})$ over the parameters, Bayesian inference aims to find the posterior $p(\boldsymbol{\theta} \mid \mathcal{D}_1, \mathcal{D}_2, \ldots, \mathcal{D}_n)$. In variational inference, we aim to find an approximation $q_{\boldsymbol{\tau}}(\boldsymbol{\theta})$ to the posterior by

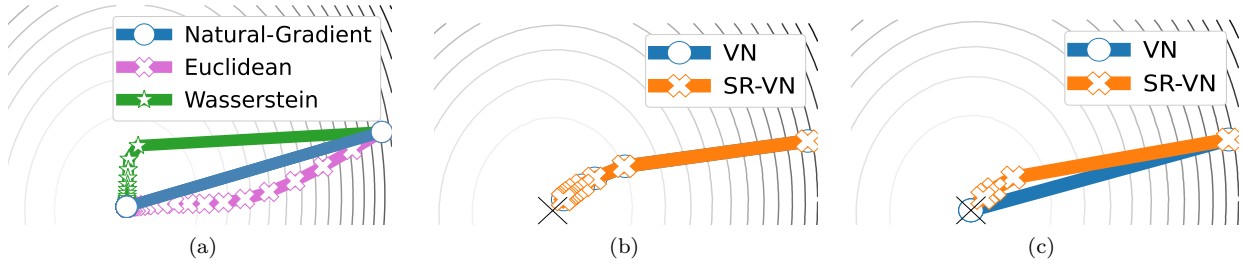

Figure 1: (a) Natural-gradient descent (circle) converges much faster (in just 1 step) than algorithms that use Euclidean (cross) or Wasserstein (star) geometries which take 10 and 470 iterations respectively. The illustration is on a 2-D Bayesian linear regression (quadratic loss in the background for all three figures). The 3 methods are taken from Khan & Rue (2023); Ranganath et al. (2014); Lambert et al. (2022b), (b) For smaller step-sizes (of the order $\rho = 10^{-5}$ or less), SR-VN and VN perform almost identically, (c) For larger step-sizes they start to differ, step-size for VN here is 1.

solving the following problem over the space $\boldsymbol{\tau} \in \Omega$ of the valid *variational parameters*,

$$\min_{\boldsymbol{\tau} \in \Omega} \ \mathbb{E}_{q_{\boldsymbol{\tau}}(\boldsymbol{\theta})}[\ell(\boldsymbol{\theta})] + \gamma \operatorname{KL}\left[q_{\boldsymbol{\tau}}(\boldsymbol{\theta}) \| p(\boldsymbol{\theta})\right], \tag{1}$$

where we denote $\ell(\boldsymbol{\theta}) = -\sum_{i=1}^{n} \log p(\mathcal{D}_i|\boldsymbol{\theta})$, $\operatorname{KL}[\cdot\|\cdot]$ to be the KL divergence, and $\gamma > 0$ is a scalar parameter commonly referred to as the temperature. When $\gamma = 1$, the objective is referred to as the negative evidence lower bound (ELBO) (Jaakkola & Jordan, 1997; Blei et al., 2017), which is also the focus of our analysis. The value of $\gamma$ plays a crucial role in shaping the posterior variance. If $\gamma$ is set too high, the KL term dominates and forces the variational posterior $q_{\boldsymbol{\tau}}(\boldsymbol{\theta})$ to collapse to the prior $p(\boldsymbol{\theta})$, eliminating any learned posterior structure. Conversely, if $\gamma$ is set too low, the KL penalty becomes negligible, which can lead to high-variance gradient estimates and unstable training (Shen et al., 2024b).

In many problems, the negative log-likelihood (NLL) $\ell(\boldsymbol{\theta})$ is convex; it could also be an arbitrary loss function, giving rise to *generalized* posterior (Zhang, 1999; Catoni, 2007). The regularizer, denoted by $R(\boldsymbol{\theta}) = -\log p(\boldsymbol{\theta})$, too is chosen to be convex. Popular examples include Bayesian linear and logistic regressions, as well as Gaussian process models. In such cases, a more convenient form is used to separately define two terms as the expected log-loss and negative-entropy respectively, that is $f(\boldsymbol{\tau}) = \mathbb{E}_{q_{\boldsymbol{\tau}}(\boldsymbol{\theta})}[\ell(\boldsymbol{\theta}) + \gamma R(\boldsymbol{\theta})]$, $\mathcal{H}(\boldsymbol{\tau}) = \mathbb{E}_{q_{\boldsymbol{\tau}}(\boldsymbol{\theta})}[\log q_{\boldsymbol{\tau}}(\boldsymbol{\theta})]$. Using these, we can rewrite Equation (1) by expanding the KL divergence as

$$\min_{\boldsymbol{\tau} \in \Omega} \mathcal{L}(\boldsymbol{\tau}) \quad \text{where} \quad \mathcal{L}(\boldsymbol{\tau}) := f(\boldsymbol{\tau}) + \gamma \mathcal{H}(\boldsymbol{\tau}), \tag{2}$$

and where the first term involves the expectation of a convex loss function while the second term is the entropy. This reformulation allows us to connect VI to a wide variety of problems in supervised, unsupervised, and reinforcement learning which can be formulated as a minimization of the form

$$\min_{\boldsymbol{\theta} \in \mathbb{R}^p} \bar{\ell}(\boldsymbol{\theta}) \quad \text{where} \quad \bar{\ell}(\boldsymbol{\theta}) := \ell(\boldsymbol{\theta}) + \gamma R(\boldsymbol{\theta}). \tag{3}$$

VI approaches can be seen as a variant of such problems where instead of finding a single minimizer $\boldsymbol{\theta}^*$, we seek a parametric distribution $q_{\boldsymbol{\tau}^*}(\boldsymbol{\theta})$ by solving Equation (2). This is closely related to techniques in robust or global optimization that use $q_{\boldsymbol{\tau}}(\boldsymbol{\theta})$ to smooth the loss and mitigate the influence of local minima (Mobahi & Fisher III, 2015; Leordeanu & Hebert, 2008; Hazan et al., 2016), often with $\gamma$ taking a value of 0. This approach is also referred to as Variational Optimization (VO) (Staines & Barber, 2012). VO can lead us to the minimum $\boldsymbol{\theta}^*$ because, for $\gamma = 0$, $\mathcal{L}(\boldsymbol{\tau})$ is an upper bound on the minimum value of $\bar{\ell}$, that is, $\min_{\boldsymbol{\theta}} \bar{\ell}(\boldsymbol{\theta}) \leq \min_{\boldsymbol{\tau} \in \Omega} \mathcal{L}(\boldsymbol{\tau})$. Therefore minimizing $\mathcal{L}(\boldsymbol{\tau})$ minimizes $\bar{\ell}(\boldsymbol{\theta})$, and when the distribution $q$ puts all its mass on $\boldsymbol{\theta}^*$, we recover the minimum value. Problem 2 with $\gamma = 0$ is also encountered across various other domains, including random search (Baba, 1981), stochastic optimization (Spall, 2005), and evolutionary strategies (Beyer, 2001). In this context, $q_{\boldsymbol{\tau}}(\boldsymbol{\theta})$ serves as the 'search' distribution employed to locate the global minimum of a black-box function, $\ell(\boldsymbol{\theta})$. Notably, in the realm of reinforcement learning, this

distribution can take on the form of a policy distribution designed to minimize the expected value-function $\ell(\boldsymbol{\theta})$ (Sutton et al., 1998), sometimes incorporating entropy regularization techniques (Williams & Peng, 1991; Teboulle, 1992; Mnih et al., 2016).

## 3 Background on NGD for VI

We will review VI using GD and NGD methods and discuss the NG flow and NGD with the Cholesky factor, and for both, we will provide convergence guarantees in the later sections.

### 3.1 Variational Gaussian Inference with GD

Throughout, we will focus on VI with a Gaussian distribution $q_{\boldsymbol{\tau}}(\boldsymbol{\theta}) := \mathcal{N}(\boldsymbol{\theta} \mid \mathbf{m}, \mathbf{V})$ where $\mathbf{m} \in \mathbb{R}^d$ is the mean, $\mathbf{V} \in \mathbb{R}^{d \times d}$ is the covariance. The variational parameter $\boldsymbol{\tau} = (\mathbf{m}, \mathbf{V})$ is only defined within the set of valid parameters $\Omega'$ – the set of $\boldsymbol{\tau}$ where $\mathbf{V}$ is positive definite, that is $\Omega' = \{(\mathbf{m}, \mathbf{V}) : \mathbf{m} \in \mathbb{R}^d, \mathbf{V} \succ 0 \in \mathbb{R}^{d \times d}\}$. The problem 2 can then be rewritten as, $\min_{\mathbf{m}, \mathbf{V}} \mathcal{L}(\mathbf{m}, \mathbf{V})$. The function $\mathcal{L}$ is differentiable under mild conditions on the interior of $\Omega'$ even when $\ell$ is not differentiable, as discussed by Staines & Barber (2012). This makes it possible to apply gradient-based optimization methods to optimize it. A straightforward approach to minimize $\mathcal{L}$ is to use Gradient Descent (GD) as shown below,

$$\mathbf{GD:} \quad \mathbf{V}_{t+1} = \mathbf{V}_t - \rho \nabla_{\mathbf{V}} \mathcal{L}(\mathbf{m}_t, \mathbf{V}_t), \quad \mathbf{m}_{t+1} = \mathbf{m}_t - \rho \nabla_{\mathbf{m}} \mathcal{L}(\mathbf{m}_t, \mathbf{V}_t), \tag{4}$$

where $\rho > 0$ is a step size. This approach is often referred to as the black-box VI (BBVI) in the literature (Ranganath et al., 2014), and offers simple and convenient updates to implement using modern automatic-differentiation methods and the reparameterization trick (Graves, 2011; Blundell et al., 2015; Titsias & Lázaro-Gredilla, 2014). Unfortunately, such direct gradient-descent methods can be slow in practice.

### 3.2 Variational Gaussian Inference with NGD

An alternative approach is to use natural-gradient descent which exploits the information geometry (Amari, 2016) of $q$ to speed up convergence. Given the Fisher information matrix (FIM) (Amari, 2008) of $q_{\boldsymbol{\tau}}(\boldsymbol{\theta})$, that is $\mathbf{F}_{\boldsymbol{\tau}} = \mathbb{E}_{q_{\boldsymbol{\tau}}(\boldsymbol{\theta})}[\nabla_{\boldsymbol{\tau}} \log q_{\boldsymbol{\tau}}(\boldsymbol{\theta})(\nabla_{\boldsymbol{\tau}} \log q_{\boldsymbol{\tau}}(\boldsymbol{\theta}))^{\top}]$, which is positive-definite for all $\boldsymbol{\tau} \in \Omega$ in our setting, the NGD for VI in the natural-parameter space (with $\boldsymbol{\tau}$ as natural parametrization) is given as follows (Khan & Rue, 2023):

$$\mathbf{NGD:} \quad \boldsymbol{\tau}_{t+1} = \boldsymbol{\tau}_t - \rho \mathbf{F}_{\boldsymbol{\tau}_t}^{-1} \nabla_{\boldsymbol{\tau}} \mathcal{L}(\boldsymbol{\tau}_t). \tag{5}$$

The preconditioning of the gradient by the FIM leads to proper scaling in each dimension which often leads to faster convergence. A notable connection between NGD for 2 and Newton's method for 3 was established by Khan et al. (2017; 2018) in the case of a Gaussian distribution with mean $\mathbf{m}$ and the precision $\mathbf{S}$, which is the inverse of the covariance $\mathbf{S} = \mathbf{V}^{-1}$. Denoting the expected gradient and the expected Hessian, respectively, as:

$$\mathbf{g}_t = \mathbb{E}_{\boldsymbol{\theta} \sim \mathcal{N}(\mathbf{m}_t, \mathbf{V}_t)}[\nabla_{\boldsymbol{\theta}} \bar{\ell}(\boldsymbol{\theta})], \quad \mathbf{H}_t = \mathbb{E}_{\boldsymbol{\theta} \sim \mathcal{N}(\mathbf{m}_t, \mathbf{V}_t)}[\nabla_{\boldsymbol{\theta}}^2 \bar{\ell}(\boldsymbol{\theta})]. \tag{6}$$

The resulting updates,

$$\mathbf{VN:} \quad \mathbf{S}_{t+1} = (1 - \gamma\rho)\mathbf{S}_t + \rho \mathbf{H}_t, \quad \mathbf{m}_{t+1} = \mathbf{m}_t - \rho(\mathbf{S}_{t+1})^{-1}\mathbf{g}_t, \tag{7}$$

referred to as the variational Newton (VN) update, optimize 2 instead of 3. The standard Newton's update for problem 3 is recovered by approximating the expectations at the mean $\mathbf{m}$ and using step-size $\rho = 1$ when $\gamma = 1$; see Khan & Rue (2023). Since the precision $\mathbf{S}$ lies in a positive-definite matrix space, the update 7 may violate this positivity constraint of the parameter space $\Omega$ (Khan & Nielsen, 2018). For example, this happens when the loss $\ell(\boldsymbol{\theta})$ is non-convex.

### 3.3 Natural Gradient Flow

The NG flow minimizes the KL divergence functional on a parameterized manifold of Gaussian densities – the objective stated in Equation (1). That is, NG flow is a continuous time update of the NGD (Equation (5),

---

**Algorithm 1 Square-Root Variational Newton (SR-VN)**

---

1: **Parameter:** Step-size $\rho > 0$ is chosen using Equation (52); tril is defined in Equation (10).
2: Initialize $(\mathbf{m}_0, \mathbf{C}_0)$ to satisfy Assumption 1
3: **for** $t = 0, \dots, T$ **do**
4:     Update $\mathbf{g}_t$ and $\mathbf{H}_t$ using Equation (6)
5:     $\mathbf{C}_{t+1} \leftarrow \mathbf{C}_t - \rho \mathbf{C}_t \, \mathrm{tril}[\mathbf{C}_t^\top \mathbf{H}_t \mathbf{C}_t - \gamma \mathbf{I}]$
6:     $\mathbf{m}_{t+1} \leftarrow \mathbf{m}_t - \rho \mathbf{C}_t \mathbf{C}_t^\top \mathbf{g}_t$
7: **end for**

---

for $\gamma = 1$) that is,

$$\frac{d\boldsymbol{\tau}_t}{dt} = -\mathbf{F}_{\boldsymbol{\tau}_t}^{-1} \nabla_{\boldsymbol{\tau}} \mathcal{L}(\boldsymbol{\tau}_t) = -\mathbf{F}_{\boldsymbol{\tau}_t}^{-1} \nabla_{\boldsymbol{\tau}} \, \mathrm{KL}\left[ q_{\boldsymbol{\tau}}(\boldsymbol{\theta}) \,||\, p(\boldsymbol{\theta} \mid \mathcal{D}) \right]. \tag{8}$$

The updates for the NG flow in $\boldsymbol{\tau} = (\mathbf{m}, \mathbf{V})$ parameterization were provided in Chen et al. (2023, Equation (4.18)) as:

$$\frac{d\mathbf{V}_t}{dt} = \mathbf{V}_t - \mathbf{V}_t \mathbf{H}_t \mathbf{V}_t, \quad \frac{d\mathbf{m}_t}{dt} = -\mathbf{V}_t \mathbf{g}_t. \tag{9}$$

Upon discretization with respect to the natural parameters, this results in the VN update with $\gamma = 1$. Refer to Appendix B for a comparison of the $\mathbf{V}$ updates, as obtained from VN. However, due to the non-convex nature of KL even with concave log-likelihood for $(\mathbf{m}, \mathbf{V})$ parameterization (see further discussion in Section 3.6), we instead focus on analyzing the NG flow in the square-root parameterization, that is $\boldsymbol{\tau} = (\mathbf{m}, \mathbf{C}) \in \Omega$, where $\Omega = \{(\mathbf{m}, \mathbf{C}) : \mathbf{m} \in \mathbb{R}^d, \mathbf{V} = \mathbf{C}\mathbf{C}^\top \succ 0 \in \mathbb{R}^{d \times d}\}$ under the assumption that $\mathbf{C}$ is square and lower-triangular with positive real diagonal entries, ensuring its invertibility. Let us define a function that takes the lower-triangular part of a matrix and halves its diagonal, that is

$$\mathrm{tril}[\mathbf{A}]_{ij} := \begin{cases} A_{ij} & i > j, \\ \frac{1}{2} A_{ii} & i = j, \\ 0 & i < j. \end{cases} \tag{10}$$

Then, to derive flow equations for the Cholesky factor, we first note from Murray (2016, Equation (7)) that the derivative of the Cholesky factor $\mathbf{C}_t$ can be obtained from the derivative of the covariance $\mathbf{V}_t = \mathbf{C}_t \mathbf{C}_t^\top$ as

$$\frac{d\mathbf{C}_t}{dt} = \mathbf{C}_t \, \mathrm{tril}[\mathbf{C}_t^{-1} \frac{d\mathbf{V}_t}{dt} \mathbf{C}_t^{-\top}], \tag{11}$$

which after using Equation (9) leads to the following flow equations:

$$\frac{d\mathbf{C}_t}{dt} = \mathbf{C}_t \, \mathrm{tril}[\mathbf{I} - \mathbf{C}_t^\top \mathbf{H}_t \mathbf{C}_t], \quad \frac{d\mathbf{m}_t}{dt} = -\mathbf{C}_t \mathbf{C}_t^\top \mathbf{g}_t. \tag{12}$$

In Theorem 1, we prove the convergence of the flow dynamics in Equation (12) under the concave log-likelihood. Importantly, the NG flow remains the same regardless of the parameterization, and therefore the flow dynamics in Equation (9) and Equation (12) trace out the same trajectory on the Gaussian manifold. Hence, we focus on square-root parametrization solely for its theoretical convenience in proving convergence.

### 3.4 Square-Root Variational Newton (SR-VN)

In this section, we introduce a different NGD update called square-root variational Newton (SR-VN), outlined in Algorithm 1, which utilizes the Cholesky factor $\mathbf{C}$ of the covariance matrix $\mathbf{V}$. SR-VN is derived through a forward Euler discretization of the NG flow defined in square-root parameterization (Equation (12)). The resultant update aligns with independently proposed updates by Tan (2025), who defined a VN-like update using the Cholesky factor, given as

$$\mathbf{C}_{t+1} = \mathbf{C}_t - \rho \mathbf{C}_t \, \mathrm{tril}[\mathbf{C}_t^\top \nabla_{\mathbf{C}} \mathcal{L}(\mathbf{m}_t, \mathbf{C}_t)], \quad \mathbf{m}_{t+1} = \mathbf{m}_t - \rho \mathbf{C}_t \mathbf{C}_t^\top \nabla_{\mathbf{m}} \mathcal{L}(\mathbf{m}_t, \mathbf{C}_t).$$

Now, as calculated in Equation (34) in Appendix A.2, we have

$$\nabla_{\mathbf{C}}\mathcal{L}(\mathbf{m}_t, \mathbf{C}_t) = (\mathbf{H}_t - \gamma(\mathbf{C}_t\mathbf{C}_t^\top)^{-1})\mathbf{C}_t. \tag{13}$$

Finally, by seeing that $\nabla_{\mathbf{m}}\mathcal{L}(\mathbf{m}_t, \mathbf{C}_t) = \mathbf{g}_t$, we can rewrite the updates from Tan (2025, Theorem 1) as

$$\textbf{SR-VN: } \mathbf{C}_{t+1} = \mathbf{C}_t - \rho\mathbf{C}_t \operatorname{tril}[\mathbf{C}_t^\top\mathbf{H}_t\mathbf{C}_t - \gamma\mathrm{I}], \quad \mathbf{m}_{t+1} = \mathbf{m}_t - \rho\mathbf{C}_t\mathbf{C}_t^\top\mathbf{g}_t, \tag{14}$$

allowing us to immediately see that these updates indeed come from a direct forward Euler discretization of Equation (12). In Theorem 2, we establish convergence guarantees for SR-VN. We also note here that other updates based on the Cholesky factor have been considered in the literature, for example, see (Sun et al., 2009; Salimbeni et al., 2018; Glasmachers et al., 2010).

### 3.5 SR-VN as an Approximation of VN

It is interesting to see that SR-VN performs a Newton-like update in $\mathbf{m}$ (Equation (14)), but does not require a matrix inversion of the preconditioner (such as the inversion of the Hessian in VN, Equation (7)). This makes it easier to implement and analyze. However, we show in Appendix B that, in reality, SR-VN computes an approximation of the inverse and is therefore only suboptimal compared to VN. Looking at Figure 1(b), we can see that when the step-size is very small, SR-VN and VN converge to the same dynamics because they both approximate the NG flow. However, the quadratic offset introduced by the inverse approximation used in SR-VN leads it to perform suboptimally compared to VN. VN, on the other hand, computes the full inverse of the FIM defined using natural parameters (Equation (5)), and it shows its superiority by achieving one-step convergence in Figure 1(c) when the step-size is larger, specifically when it is set to 1.

### 3.6 Variational Parameters and Convexity

NGD updates are often written in the natural-parameter space 5, but this can destroy the convexity of the problem. Specifically, given a convex function $\bar{\ell}(\boldsymbol{\theta})$, it is well-known that $\mathbb{E}_{\boldsymbol{\theta} \sim \mathcal{N}(\mathbf{m}, \mathbf{V})}[\bar{\ell}(\boldsymbol{\theta})]$ is jointly convex w.r.t. $\mathbf{m}$ and the Cholesky factor $\mathbf{C}$ of $\mathbf{V}$, but not w.r.t $\mathbf{V}$ or its inverse (Challis & Barber, 2013; Domke, 2019). That is, even for the simplest convex cases, such as logistic regression, the convexity is lost in the natural and expectation parameter spaces. For example, given $n$ training data points $\{\mathbf{x}_i, y_i\}_{i=1}^n$ with $\mathbf{x}_i \in \mathbb{R}^d$ and $y_i \in \{-1, +1\}$, in Bayesian logistic regression we minimize the following loss that contains the $\ell_2$-regularization with regularization constant $\beta > 0$:

$$\bar{\ell}_{lg}(\boldsymbol{\theta}) = \sum_{i=1}^n \left[ \log(1 + \exp\{y_i(\boldsymbol{\theta}^\top\mathbf{x}_i)\}) \right] + \frac{\beta}{2}\|\boldsymbol{\theta}\|^2 \tag{15}$$

The Hessian takes the form $\nabla_{\boldsymbol{\theta}}^2\bar{\ell}_{lg}(\boldsymbol{\theta}) = \mathbf{X}\mathbf{D}\mathbf{X}^\top + \beta\mathrm{I}$, where $\mathbf{X} = [\mathbf{x}_1, \ldots, \mathbf{x}_n]^\top \in \mathbb{R}^{n \times d}$ is the data matrix, $\mathbf{D} \in \mathbb{R}^{d \times d}$ is a diagonal matrix such that $D_{ii} = \boldsymbol{\sigma}(\boldsymbol{\theta}^\top\mathbf{x}_i)(1 - \boldsymbol{\sigma}(\boldsymbol{\theta}^\top\mathbf{x}_i))$ with the sigmoid function $\boldsymbol{\sigma}(z) = 1/(1 + e^{-z})$. Given this form of the Hessian, it is clear that the Hessian is positive-definite (and hence regularized logistic loss is $\beta$-strongly convex), however, the expected loss $\mathbb{E}_{\boldsymbol{\theta} \sim q_\tau(\boldsymbol{\theta})}[\bar{\ell}_{lg}(\boldsymbol{\theta})]$ is non-convex in natural and expectation parameterizations, or even in the $(\mathbf{m}, \mathbf{V})$ parameterization of the Gaussian.

This issue makes it harder to analyze convergence guarantees of NGD in Equation (5) in general, and therefore for the algorithm given in Equation (7). The same is true for the NG flow in Equation (9), because KL is non-convex in $\boldsymbol{\tau} = (\mathbf{m}, \mathbf{V})$. The issue is specifically noted by Chérief-Abdellatif et al. (2019) for NGD. They analyze the convergence of online VI by using tools from convex optimization, but their proof techniques do not extend to NGD because of the loss of convexity in the natural or expectation parameters. Our goal in this paper is to address this challenge. Our key idea is to simply switch to the square-root parameterization where the convexity properties are preserved. We consider specific NGD variants that use such square-root parameterization and derive strong convergence guarantees for them. Additionally, by leveraging the strong convexity of the log-likelihood we show that KL functional satisfies a Riemannian PL inequality in square-root parameterization, which eventually leads us to prove convergence for NG flow.

## 4 Assumptions

We will use the following assumptions for our analysis.

**Assumption 1** (Initialization). *We initialize Algorithm 1 with $\boldsymbol{\tau}_0 = (\mathbf{m}_0, \mathbf{C}_0) = (\mathbf{m}_0, C_0 \mathrm{I})$ for a constant $C_0 > 0$ and $\mathbf{m}_0 \in \mathbb{R}^d$.*

**Assumption 2** (Strong Convexity). *We assume that the negative log-likelihood $\bar{\ell}(\boldsymbol{\theta})$ is $\delta$-strongly-convex in $\boldsymbol{\theta}$, that is, $\nabla^2_{\boldsymbol{\theta}} \bar{\ell}(\boldsymbol{\theta}) \succcurlyeq \delta \mathrm{I}$. Then, from (Domke, 2019, Theorem 9) $\mathcal{L}(\mathbf{m}, \mathbf{C})$ is $\delta$-strongly-convex in square-root parametrization $\boldsymbol{\tau} = (\mathbf{m}, \mathbf{C})$, that is*

$$\mathcal{L}(\boldsymbol{\tau}') \geq \mathcal{L}(\boldsymbol{\tau}) + \langle \nabla_{\boldsymbol{\tau}} \mathcal{L}(\boldsymbol{\tau}), \boldsymbol{\tau}' - \boldsymbol{\tau} \rangle + \frac{\delta}{2} \left\| \boldsymbol{\tau}' - \boldsymbol{\tau} \right\|^2 \tag{16}$$

**Assumption 3** (Lipschitz Gradient). *We assume that the negative log-likelihood $\bar{\ell}(\boldsymbol{\theta})$ is $M$-Lipschitz-smooth in $\boldsymbol{\theta}$, that is, $\nabla^2_{\boldsymbol{\theta}} \bar{\ell}(\boldsymbol{\theta}) \preccurlyeq M \mathrm{I}$. Then, from (Domke, 2020, Theorem 1) $\mathcal{L}(\mathbf{m}, \mathbf{C})$ is $M$-Lipschitz-smooth in square-root parametrization $\boldsymbol{\tau} = (\mathbf{m}, \mathbf{C})$, that is*

$$\mathcal{L}(\boldsymbol{\tau}') \leq \mathcal{L}(\boldsymbol{\tau}) + \langle \nabla_{\boldsymbol{\tau}} \mathcal{L}(\boldsymbol{\tau}), \boldsymbol{\tau}' - \boldsymbol{\tau} \rangle + \frac{M}{2} \left\| \boldsymbol{\tau}' - \boldsymbol{\tau} \right\|^2 \tag{17}$$

**Assumption 4** (Bounded Iterates). *Let the iterates generated at time $t \in \mathbb{N}_0$ from Algorithm 1 be $\mathbf{C}_t$. Then the following two statements are true.*

1. *There exist constants $0 < \xi_l \leq \xi_u < \infty$ such that $\xi_l \leq \|\mathbf{C}_t\|_F \leq \xi_u$ for all $t \geq 0$.*

2. *There exists a constant $\lambda_{\min} > 0$ such that $\mathbf{V}_t \succcurlyeq \lambda_{\min} \mathrm{I}$ for all $t \geq 0$.*

*Remark* 1. The largest singular value of $\mathbf{V}_t$ (that is, $\|\mathbf{V}_t\|$) can be upper bounded using Assumption 4 as follows: $\|\mathbf{V}_t\| = \left\| \mathbf{C}_t \mathbf{C}_t^\top \right\| \leq \|\mathbf{C}_t\|^2 \leq \|\mathbf{C}_t\|_F^2 \leq \xi_u^2$.

Assumptions 2 and 3 hold for commonly encountered loss functions, e.g. the $\ell_2$-regularized logistic loss 15. Assumption 4 ensures that the updates remain bounded throughout optimization, and enforces the positive-definiteness of $\mathbf{V}$.

**The assumptions hold for Logistic Regression.** We saw in Section 3.6 that the Hessian of the logistic loss takes the form $\mathbf{X} \mathbf{D} \mathbf{X}^\top + \beta \mathrm{I}$, where the diagonal matrix $\mathbf{D}$ has entries bounded between 0 and $1/4$. Hence, the Hessian of the regularized logistic loss is bounded as long as the data matrix $\mathbf{X}$ is bounded (which is often the case), i.e., $\beta \mathrm{I} \preccurlyeq \mathbf{H} \preccurlyeq \zeta \mathrm{I}$ where $\zeta = \lambda_{\max}(\mathbf{X} \mathbf{D} \mathbf{X}^\top) + \beta$. Thus, the assumption of a bounded square root in part 1 of Assumption 4 holds with an appropriate selection of step-size in Equation (14). Additionally, Assumption 2 and part 2 of Assumption 4 are met because regularized logistic regression is $\beta$-strongly convex. Therefore, Assumptions 1–4 apply to the original VN's $\mathbf{V}$ update when working with the regularized logistic loss. As confirmed empirically, the same conclusion extends to the SR-VN's $\mathbf{C}$ update.

## 5 Analysis

In this section, we will analyze the convergence guarantees for the NG flow (Equation (12)) and SR-VN (Algorithm 1). We will do this by applying tools from convex optimization, considering a set of assumptions related to boundedness and convexity. To prove convergence, we need Assumption 2 and part 2 of Assumption 4 for the NG flow, and Assumptions 1–4 for SR-VN.

### 5.1 Flow Convergence with Riemannian PL

We first show that, under the assumption of strongly convex NLL (Assumption 2), the KL functional satisfies a local Riemannian PL inequality in the square-root parameterization (refer to Lemma 2). This result paves the way for proving the convergence of the NG flow, as outlined in Theorem 1.

We first state the following lemma that ensures that the FIM is bounded, and thus well-behaved, which is crucial for our entire analysis. The detailed proof for all the stated results can be found in Appendix C.

**Lemma 1** (Bounded FIM eigenvalues). *For $\boldsymbol{\tau} = (\mathbf{m}, \mathbf{C})$, given Assumptions 4, we conclude that*

$$\lambda_{\min}^g \begin{pmatrix} \mathbf{I}_{d \times d} & 0 \\ 0 & \mathbf{I}_{d^2 \times d^2} \end{pmatrix} \preccurlyeq \mathbf{F}_{\boldsymbol{\tau}}^{-1} \preccurlyeq \lambda_{\max}^g \begin{pmatrix} \mathbf{I}_{d \times d} & 0 \\ 0 & \mathbf{I}_{d^2 \times d^2} \end{pmatrix},$$

*where*

$$\lambda_{\min}^g = \min\left\{ \lambda_{\min}, \frac{\lambda_{\min}^2}{2} \right\}, \quad and \quad \lambda_{\max}^g = \xi_u^2. \tag{18}$$

We know that any minimizer $q_{\boldsymbol{\tau}_*}(\boldsymbol{\theta}) = \mathcal{N}(\boldsymbol{\theta} | \mathbf{m}_*, \mathbf{V}_*)$ satisfies (Khan & Rue, 2023; Lambert et al., 2022a)

$$\mathbb{E}_{\boldsymbol{\theta} \sim q_{\boldsymbol{\tau}_*}(\boldsymbol{\theta})}[\nabla_{\boldsymbol{\theta}} \bar{\ell}(\boldsymbol{\theta})] = 0, \quad and \quad \mathbf{V}_*^{-1} = \mathbb{E}_{\boldsymbol{\theta} \sim q_{\boldsymbol{\tau}_*}(\boldsymbol{\theta})}[\nabla_{\boldsymbol{\theta}}^2 \bar{\ell}(\boldsymbol{\theta})]. \tag{19}$$

**Existence of limit points.** The limit points $\mathbf{m}^*, \mathbf{C}^*$ exist because of the strong convexity assumption (Assumption 2) which implies there exists a $\boldsymbol{\tau}^* = (\mathbf{m}^*, \mathbf{C}^*)$ that uniquely minimizes $\mathcal{L}$. The existence of this $\boldsymbol{\tau}^*$ can be argued using the fact that $\mathcal{L}$ is continuous and coercive and $\Omega$ (the set of $\boldsymbol{\tau}'$s) can be extended to a closed set (but the minimizer can be shown to lie inside $\Omega$ due to Equation (19)). Then, a global minimum exists due to the Weierstrass extreme value theorem.

As a consequence of Lemma 1, one can prove the following lemma, given that NLL is strongly-convex, which establishes a PL inequality where we measure the magnitude of the gradient in the (natural) Riemannian geometry. This will be needed to prove the convergence guarantees for the Riemannian gradient flow and descent.

**Lemma 2** (Riemannian PL Inequality). *The KL functional satisfies a local Polyak-Łojasiewicz (PL) inequality with PL constant $\mu = \delta \lambda_{\min}^g$, that is,*

$$\left\| \nabla_{\boldsymbol{\tau}} \mathrm{KL}\left[ q_{\boldsymbol{\tau}}(\boldsymbol{\theta}) \,||\, p(\boldsymbol{\theta} \mid \mathcal{D}) \right] \right\|_{\mathbf{F}_{\boldsymbol{\tau}}^{-1}}^2 \geq 2\mu \left( \mathrm{KL}\left[ q_{\boldsymbol{\tau}}(\boldsymbol{\theta}) \,||\, p(\boldsymbol{\theta} \mid \mathcal{D}) \right] - \mathrm{KL}\left[ q_{\boldsymbol{\tau}_*}(\boldsymbol{\theta}) \,||\, p(\boldsymbol{\theta} \mid \mathcal{D}) \right] \right). \tag{20}$$

*Remark* 2. Lemma 2 is a 'local' PL inequality because it only holds for the chosen parametrization. This is because of the use of strong-convexity of ELBO for its proof (see Appendix C.2), which only holds for $(\mathbf{m}, \mathbf{C})$ parametrization. Since one does not know if convexity will hold in all parametrizations there is no guarantee that the PL inequality will hold globally.

Once the Riemannian PL condition is established, we can derive a convergence rate using a standard Lyapunov function analysis. As mentioned earlier, this analysis can be modified to incorporate a bias term as shown in Appendix D.

**Theorem 1.** *Under Assumptions 2 and 4, the NG flow dynamics defined in Equation* (12) *satisfy*

$$\mathrm{KL}\left[ q_{\boldsymbol{\tau}_t}(\boldsymbol{\theta}) \,||\, p(\boldsymbol{\theta} \mid \mathcal{D}) \right] - \mathrm{KL}\left[ q_{\boldsymbol{\tau}_*}(\boldsymbol{\theta}) \,||\, p(\boldsymbol{\theta} \mid \mathcal{D}) \right]$$

$$\leq e^{-2\mu t} \left( \mathrm{KL}\left[ q_{\boldsymbol{\tau}_0}(\boldsymbol{\theta}) \,||\, p(\boldsymbol{\theta} \mid \mathcal{D}) \right] - \mathrm{KL}\left[ q_{\boldsymbol{\tau}_*}(\boldsymbol{\theta}) \,||\, p(\boldsymbol{\theta} \mid \mathcal{D}) \right] \right).$$

### 5.2 Convergence of SR-VN

Under Assumption 1–2, we give what we believe are the first rigorous convergence guarantees for optimizing 2 using Algorithm 1. Specifically, we show an exponential rate of convergence for Algorithm 1, as stated in Theorem 2, with a detailed proof provided in Appendix C.4. The literature has suggested various updates, as seen in references Lin et al. (2019; 2021b;a); Khan & Rue (2023). It is also worth highlighting that our proof extends to generic NGD algorithms that use *any* square-root parametrization. The proof technique used to derive Theorem 2 can easily be adapted to prove convergence for these other variants and is omitted here to avoid redundancy.

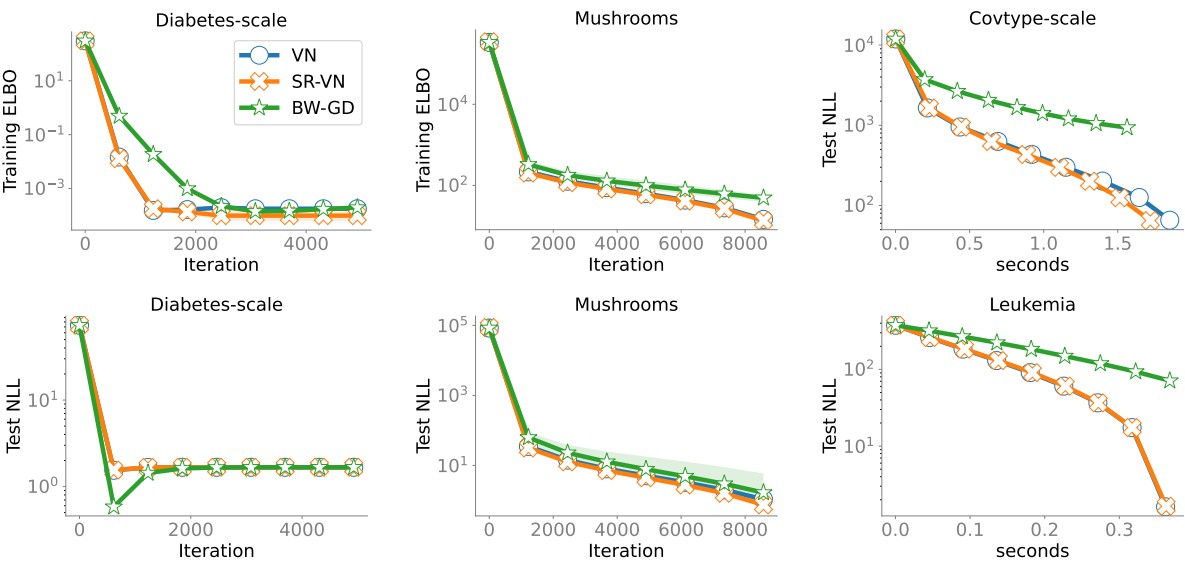

Figure 2: For small-scale LIBSVM datasets, we show the training ELBO and test NLL w.r.t number of iterations (left and middle panel) and time (right panel). The min and max of the plotted values are displayed around their averages (taken over five random initializations). We also subtracted the min values achieved (for both training and testing) over all iterations from all three methods and only plotted the resulting values. We can see that, in most cases, SR-VN and VN perform similarly while BW-GD tends to be slower. However, sometimes SR-VN could also be slower than the two, for example, see Appendix F.3.

**Theorem 2** (Global Convergence). *Given Assumptions 1–4 and considering the limit points $\mathbf{m}_*$ and $\mathbf{C}_*$ in Equation* (19)*, the updates provided by Algorithm 1 satisfy*

$$\mathcal{L}(\mathbf{m}_{t+1}, \mathbf{C}_{t+1}) - \mathcal{L}(\mathbf{m}_*, \mathbf{C}_*) \leq (1 - 2\eta\delta)^{t+1}(\mathcal{L}(\mathbf{m}_0, \mathbf{C}_0) - \mathcal{L}(\mathbf{m}_*, \mathbf{C}_*)), \tag{21}$$

*where*

$$\eta = \min\left\{\lambda_{\min}\rho - \frac{M\rho^2\xi_u^4}{2}, \quad \sqrt{\frac{5}{2}}\rho\xi_l^2 - \frac{5\rho^2\xi_u^4 M}{4}\right\}. \tag{22}$$

*For convergence, we require that $0 < (1 - 2\eta\delta) < 1$. This constraints the step-size $\rho > 0$ to satisfy certain conditions. The permissible step-sizes satisfying the above constraint are discussed in Appendix C.5, along with the time complexity of SR-VN in Equation* (55)*.*

## 6 Experiments

In this section, we present results on the Bayesian logistic regression problem introduced in 15 to optimize 2, where we compare SR-VN to two other methods, namely 1) BW-GD Lambert et al. (2022b, Algorithm 1) and 2) Variational Newton in Equation (7). The BW-GD updates are given as:

$$\text{BW-GD:} \quad \mathbf{m}_{t+1} = \mathbf{m}_t - \alpha \, \mathbb{E}_{\mathcal{N}(\mathbf{m}_t, \mathbf{V}_t)}[\nabla_{\boldsymbol{\theta}}\bar{\ell}(\boldsymbol{\theta})],$$

$$\mathbf{M}_t = \mathbf{I} - \alpha \left(\mathbb{E}_{\mathcal{N}(\mathbf{m}_t, \mathbf{V}_t)}[\nabla_{\boldsymbol{\theta}}^2\bar{\ell}(\boldsymbol{\theta})] - \mathbf{S}_t\right), \quad \mathbf{V}_{t+1} = \mathbf{M}_t\mathbf{V}_t\mathbf{M}_t$$

where $\alpha > 0$ is the step-size. Due to space constraints, details of all the datasets used and the setting of various algorithmic parameters for these methods are given in Appendix F. The additional results and experiments are provided in Appendix F.2.

### 6.1 Bayesian logistic regression

We run our experiments on both small-scale and large-scale datasets to compare the aforementioned methods. The way we measure the performance of methods is via two metrics: the negative ELBO on the training

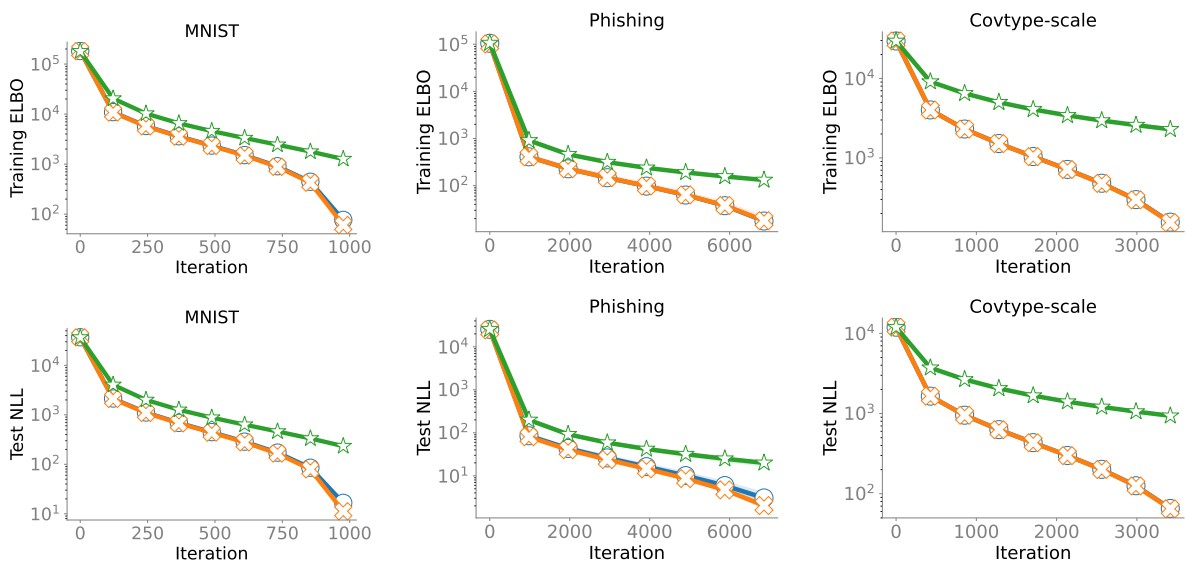

Figure 3: Comparison on Large-Scale Datasets. The same trends hold at scale: both SR-VN and VN exhibit comparable performance, whereas BW-GD tends to be slower.

dataset and the NLL on the test dataset. We report the average of the NLL over all test points, i.e. we compare the NLL on the test set computed as follows: $-\sum_{i=1}^{n_{\text{test}}} \log(1 + \exp\{-y_i(\hat{\boldsymbol{\theta}}^\top \mathbf{x}_i)\})/n_{\text{test}}$ where $\hat{\boldsymbol{\theta}}$ is parameter estimate and $n_{\text{test}}$ is the number of examples in the test set.

**Datasets.** We consider eight different LIBSVM datasets[1] (Chang & Lin, 2011), consisting of five small and three large-scale datasets. The description of these datasets is provided in Table 2 of Appendix F. Here, we show results for two small-scale datasets (see Figure 2), namely Diabetes-scale ($n = 768, d = 8, n_{\text{train}} = 614$) and Mushrooms ($n = 8124, d = 112, n_{\text{train}} = 64, 99$). For large-scale datasets (see Figure 3), we show MNIST ($n = 70, 000, d = 784, n_{\text{train}} = 60, 000$), Covtype-scale ($n = 581, 012, d = 54, n_{\text{train}} = 500, 000$), and Phishing ($n = 11, 055, d = 68, n_{\text{train}} = 8, 844$) datasets. All these datasets fall in the case where $n > d$. For the $d > n$ case, we choose the Leukemia dataset ($n = 38, d = 7, 129, n_{\text{train}} = 34$) and show time plots to be inclusive of all possible different settings. Note that different datasets use binary labels other than $\{-1, +1\}$, we mapped them all to the same labels so that loss evaluation is uniform across all datasets. All experiments are performed on NVIDIA GeForce RTX 3090 GPUs.

We can observe that in the majority of instances, SR-VN performs similarly to VN and both of them are always better than BW-GD. Most often, VN is slightly better than SR-VN, indicating its superiority, and motivating its analysis as a future work. As a side note, one can observe that the wall-clock time shoots up for SR-VN in the rightmost panel of Figure 2. The reason for this lies in the difference between the computational complexity of algorithms presented, this is discussed in more detail in Appendix F.1.

## 7 Related Works

**NGVI.** NGVI approximates intractable posterior distributions by employing natural gradients, which consider the geometry of the parameter space, instead of standard gradients to update the variational parameters (Hoffman et al., 2013). This approach is believed to lead to more robust and efficient optimization (Amari, 1998), especially when dealing with complex models and high-dimensional data (Shen et al., 2024a). Deterministic NGVI (Honkela et al., 2008; 2010; Godichon-Baggioni et al., 2024), in contrast to its stochastic counterpart, computes the exact natural gradient at each iteration. This can be computationally more demanding but can also lead to more accurate and stable updates. When feasible, the benefit of this approach is that one can use a constant step-size.

---

[1]Available at `https://www.csie.ntu.edu.tw/~cjlin/libsvmtools/datasets/`

**Convergence Analysis.** While NGVI has shown promising empirical performance—for example, in latent Dirichlet allocation topic models (Hoffman et al., 2013), Bayesian neural networks(Khan & Nielsen, 2018; Khan et al., 2018; Osawa et al., 2019; Shen et al., 2024a), probabilistic graphical models (Johnson et al., 2016), and large-scale Gaussian processes (Hensman et al., 2013; 2015; Salimbeni et al., 2018)—the theoretical analysis of its convergence rates is still an active area of research (Theis & Hoffman, 2015). Existing works often focus on specific model classes or makes simplifying assumptions to derive convergence guarantees. For instance, (Wu & Gardner, 2024) showed that stochastic NGVI, exhibits a non-asymptotic convergence rate of $\mathcal{O}(1/T)$ for conjugate models, where $T$ represents the number of iterations. This rate is comparable to SGD, but stochastic NGVI often has has better constants in the convergence rate, leading to faster convergence in practice (Khan & Nielsen, 2018). However, for non-conjugate models, stochastic NGVI with canonical parameterization operates on a non-convex objective, making it challenging to establish a global convergence rate. In the context of Gaussian process models, (Tang & Ranganath, 2019) demonstrated the effectiveness of natural gradients in both conjugate and non-conjugate scenarios, showing potential for faster convergence compared to traditional gradient-based methods.

Despite these advancements, a comprehensive understanding of the convergence behavior of NGVI in general settings remains an open question (Regier et al., 2017). This is partly due to the difficulty in analyzing NGD for commonly-used parameterizations that can destroy the concavity of the log-likelihood (Chérief-Abdellatif, 2020). Our work addressed this issue by considering a square-root parameterization for variational Gaussian inference, which helps maintain the concavity of the log-likelihood and facilitates the convergence analysis. We proved an exponential convergence rate for both NG flow and NGD under this parameterization, bridging the gap between theory and practice for NGD convergence guarantees. Similar efforts to analyze NGD convergence have been made by Khan et al. (2016) for general stochastic, non-convex settings and (Chen et al., 2023) for continuous-time flow with concave log-likelihoods. However, these studies either have slow convergence rates or focus on restrictive settings, leaving room for improvement in more general cases.

**Parameterization Choices.** The choice of parameterization plays a crucial role in the effectiveness of NGVI (Lin et al., 2019). Different parameterizations can lead to different geometries in the parameter space, affecting the trajectory of the optimization process. Studies have explored various parameterizations for NGVI, including natural and expectation parameters (Honkela et al., 2010; Khan & Rue, 2023; Godichon-Baggioni et al., 2024). Ko et al. (2024) proposed an alternative approach based on a score-based divergence that can be optimized by a closed-form proximal update for Gaussian variational families with full covariance matrices. They proved that in the limit of infinite batch size, the variational parameter updates converge exponentially quickly to the target mean and covariance. Our work focused on square-root parameterizations for variational Gaussian inference, extending the idea introduced by Tan (2025) who utilized the Cholesky factor. This parameterization helps maintain the concavity of log-likelihood functions, which in turn facilitates convergence guarantees for both NG flow and NGD. While this approach may lead to suboptimal convergence in some cases, it provides a valuable framework for analyzing and understanding the convergence behavior of deterministic NGVI.

## 8 Discussion and Conclusion

In this paper, we studied the performance of NG flow and NGD for optimizing the ELBO, a popular objective function in VI. Our findings suggest that the effectiveness of these methods can be affected by the choice of parameterization and the underlying geometry of the optimization landscape; however, a definitive link has yet to be established. Motivated by the need for a principled convergence analysis, we focused on a square-root parameterization for variational Gaussian inference. This approach, building upon the work of Domke (2019), has the promise of preserving the concavity of the log-likelihood, a property often lost in traditional parameterizations. By preserving concavity, we were able to establish theoretical guarantees for the convergence of both NG flow and NGD under this parameterization, thus bridging a gap between theoretical analysis and practical implementation of NGD.

Our empirical evaluations further validated the benefits of the square-root parameterization. We observed that it generally performs comparably to the standard natural parameterization, exhibiting fast convergence due to its Newton-like update. Moreover, our experiments, particularly the toy (ill-conditioned) problem in

Figure 1, highlighted the role of the square-root parameterization in maintaining stable updates, especially in scenarios where traditional parameterizations might be sensitive to parameter choices or errors in the Fisher information matrix estimation.

Despite the advancements, challenges and open questions remain. Our findings pave the way for future research in several key directions, this includes extending our analysis to stochastic NGVI—common in real-world settings—to study how noise influences optimization and whether the square-root parameterization remains effective, along with the validity of key assumptions (e.g., Assumptions 2-4) for more complex applications like deep learning.

### Broader Impact Statement

Our work makes several contributions: i) we propose a theoretical analysis using a square-root parameterization for the Gaussian covariance, which we expect can encourage further research into establishing convergence guarantees for natural-gradient variational inference, particularly for cases involving concave log-likelihoods. ii) Our experimental results reveal that natural gradient methods outperform algorithms relying on Euclidean or Wasserstein geometries, showcasing their practical advantages in terms of convergence speed and effectiveness, which could impact the development of more robust inference algorithms in probabilistic modeling.

### Acknowledgments

Navish Kumar and Aurelien Lucchi acknowledge the financial support of the Swiss National Foundation, SNF grant No 207392.

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

**Table of Contents**

# A    Gradient Calculations

This section focuses on showing calculations that lead to the gradients used to prove results in the main paper. In Appendix D in Khan & Rue (2023), the gradients used to define VN were calculated using the moment parametrization (which we will define in a bit) $\boldsymbol{\tau} = \boldsymbol{\mu} = (\boldsymbol{\mu}_1, \boldsymbol{\mu}_2)$, i.e. gradient of the objective $\mathcal{L}(\boldsymbol{\mu})$ were calculated under this parameterization. The goal of this section is to use these gradients to derive the gradients w.r.t $\boldsymbol{\tau} = (\mathbf{m}, \mathbf{V})$ and $\boldsymbol{\tau} = (\mathbf{m}, \mathbf{C})$ parameterizations.

Before we begin, it is useful to learn about the two important parameterizations for the Exponential family. The first one is called *moments* or *mean/expectation parameters*, and are given as:

$$\boldsymbol{\mu}_1 = \mathbf{m} \in \mathbb{R}^d, \ \boldsymbol{\mu}_2 = \mathbf{S}^{-1} + \mathbf{m}\mathbf{m}^\top \in \mathbb{S}_{++}^d, \tag{23}$$

where $\mathbb{S}_{++}^d = \{\mathbf{X} \in \mathbb{R}^{d \times d} : \mathbf{X} = \mathbf{X}^\top, \mathbf{X} \succ 0\}$ is the set of symmetric positive-definite matrices. The second parametrization is given via the *natural parameters* as

$$\boldsymbol{\lambda}_1 = \mathbf{S}\mathbf{m} \in \mathbb{R}^d, \ \boldsymbol{\lambda}_2 = -\frac{1}{2}\mathbf{S} \in \mathbb{S}_{++}^d, \tag{24}$$

The way these two parameterizations are interrelated is via the following relation

$$\boldsymbol{\mu} = \nabla_{\boldsymbol{\lambda}}\mathcal{H}^*(\boldsymbol{\lambda}) \iff \nabla_{\boldsymbol{\mu}}\mathcal{H}(\boldsymbol{\mu}) = \boldsymbol{\lambda}, \tag{25}$$

where $\mathcal{H}^*(\boldsymbol{\lambda})$ is a convex function, and its *convex conjugate* $\mathcal{H}(\boldsymbol{\mu})$ returns the *entropy*[2] of the Gaussian distribution given its moments $\boldsymbol{\mu}$. These details are explained in Chapter 3 in Wainwright et al. (2008), see also the paper by Malagò & Pistone (2015).

## A.1    Gradients in $(\mathbf{m}, \mathbf{V})$ Parametrization

Now let us look at the derivatives of $f$ w.r.t $\boldsymbol{\mu}_1$ and $\boldsymbol{\mu}_2$, but first we note that:

$$\nabla_{\boldsymbol{\mu}_1}\mathcal{H}(\boldsymbol{\mu}) = \boldsymbol{\lambda}_1 \overset{(25),(24)}{=} \mathbf{S}\mathbf{m}, \tag{26}$$

Let us define

$$\mathbf{g}(\boldsymbol{\tau}) := \mathbb{E}_{\boldsymbol{\theta} \sim q_{\boldsymbol{\tau}}}[\nabla_{\boldsymbol{\theta}}\bar{\ell}(\boldsymbol{\theta})], \quad \mathbf{H}(\boldsymbol{\tau}) := \mathbb{E}_{\boldsymbol{\theta} \sim q_{\boldsymbol{\tau}}}[\nabla_{\boldsymbol{\theta}}^2\bar{\ell}(\boldsymbol{\theta})],$$

where $\boldsymbol{\tau} = (\mathbf{m}, \mathbf{V})$ when $q_{\boldsymbol{\tau}} = \mathcal{N}(\mathbf{m}, \mathbf{V})$, and when this is the case, we omit writing $\mathbf{g}(\mathbf{m}, \mathbf{V}), \mathbf{H}(\mathbf{m}, \mathbf{V})$ in favour of only writing $\mathbf{g}, \mathbf{H}$. Then, in Equation 10 in Khan & Rue (2023), it is given that:

$$\nabla_{\boldsymbol{\mu}_1}f(\boldsymbol{\mu}) = \mathbb{E}_{\boldsymbol{\theta} \sim \mathcal{N}(\mathbf{m}, \mathbf{V})}[\nabla_{\boldsymbol{\theta}}\bar{\ell}(\boldsymbol{\theta})] - \mathbb{E}_{\boldsymbol{\theta} \sim \mathcal{N}(\mathbf{m}, \mathbf{V})}[\nabla_{\boldsymbol{\theta}}^2\bar{\ell}(\boldsymbol{\theta})]\mathbf{m} = \mathbf{g} - \mathbf{H}\mathbf{m}. \tag{27}$$

Combining the above two equations, we get:

$$\begin{aligned}
\nabla_{\boldsymbol{\mu}_1}\mathcal{L}(\boldsymbol{\mu}) &= \nabla_{\boldsymbol{\mu}_1}\left(\mathbb{E}_{\boldsymbol{\theta} \sim \mathcal{N}(\mathbf{m}, \mathbf{V})}[\bar{\ell}(\boldsymbol{\theta})] + \gamma\mathcal{H}(\boldsymbol{\mu})\right) \\
&= \nabla_{\boldsymbol{\mu}_1}\mathbb{E}_{\boldsymbol{\theta} \sim \mathcal{N}(\mathbf{m}, \mathbf{V})}[\bar{\ell}(\boldsymbol{\theta})] + \gamma\nabla_{\boldsymbol{\mu}_1}\mathcal{H}(\boldsymbol{\mu}) \\
&= \mathbf{g} - \mathbf{H}\mathbf{m} + \gamma\mathbf{S}\mathbf{m},
\end{aligned} \tag{28}$$

and from Equation 11 in Khan & Rue (2023), we have

$$\nabla_{\boldsymbol{\mu}_2}\mathcal{L}(\boldsymbol{\mu}) = \frac{1}{2}(\mathbf{H} - \gamma\mathbf{S}) \tag{29}$$

---

[2]We take $\mathcal{H}(\boldsymbol{\mu})$ as the negative entropy in this paper.

This further leads to

$$
\begin{aligned}
(\nabla_{\mathbf{m}}\mathcal{L}(\boldsymbol{\mu}))_i = \frac{\partial\mathcal{L}(\boldsymbol{\mu})}{\partial m_i} &= \frac{\partial\mathcal{L}}{\partial\boldsymbol{\mu}_1}^\top \frac{\partial\boldsymbol{\mu}_1}{\partial m_i} + \operatorname{tr}\left(\frac{\partial\mathcal{L}}{\partial\boldsymbol{\mu}_2}^\top \frac{\partial\boldsymbol{\mu}_2}{\partial m_i}\right) \\
&= \left(\frac{\partial\mathcal{L}}{\partial\boldsymbol{\mu}_1}\right)_i^\top + \sum_{k,l}\left(\frac{\partial\mathcal{L}}{\partial\boldsymbol{\mu}_2}^\top\right)_{kl}\left(\frac{\partial\boldsymbol{\mu}_2}{\partial m_i}\right)_{lk} \\
&= \left(\frac{\partial\mathcal{L}}{\partial\boldsymbol{\mu}_1}\right)_i^\top + \sum_{k\neq i}\left(\frac{\partial\mathcal{L}}{\partial\boldsymbol{\mu}_2}^\top\right)_{ki} m_k + \sum_{l\neq i}\left(\frac{\partial\mathcal{L}}{\partial\boldsymbol{\mu}_2}^\top\right)_{il} m_l + 2\left(\frac{\partial\mathcal{L}}{\partial\boldsymbol{\mu}_2}^\top\right)_{ii} m_i \\
&= \left(\frac{\partial\mathcal{L}}{\partial\boldsymbol{\mu}_1}\right)_i^\top + 2\sum_{k\neq i}\left(\frac{\partial\mathcal{L}}{\partial\boldsymbol{\mu}_2}^\top\right)_{ki} m_k + 2\left(\frac{\partial\mathcal{L}}{\partial\boldsymbol{\mu}_2}^\top\right)_{ii} m_i \\
&= (\nabla_{\boldsymbol{\mu}_1}\mathcal{L})_i^\top + 2\underbrace{(\nabla_{\boldsymbol{\mu}_2}\mathcal{L}\mathbf{m})_i}_{i\text{-th row}}
\end{aligned}
$$

$$
\begin{aligned}
\Rightarrow \nabla_{\mathbf{m}}\mathcal{L} &= \nabla_{\boldsymbol{\mu}_1}\mathcal{L} + 2\nabla_{\boldsymbol{\mu}_2}\mathcal{L}\mathbf{m} \\
&\overset{(28),(29)}{=} \left(\mathbf{g} - \mathbf{Hm} + \gamma\mathbf{S}\mathbf{m}_t\right) + 2\left(\frac{1}{2}(\mathbf{H} - \gamma\mathbf{S})^\top\mathbf{m}\right) \\
&= \mathbf{g}.
\end{aligned}
\tag{30}
$$

Similarly,

$$
\begin{aligned}
(\nabla_{\mathbf{V}}\mathcal{L}(\boldsymbol{\mu}))_{ij} = \frac{\partial\mathcal{L}(\boldsymbol{\mu})}{\partial V_{ij}} &= \frac{\partial\mathcal{L}}{\partial\boldsymbol{\mu}_1}^\top \frac{\partial\boldsymbol{\mu}_1}{\partial V_{ij}} + \operatorname{tr}\left(\frac{\partial\mathcal{L}}{\partial\boldsymbol{\mu}_2}^\top \frac{\partial\boldsymbol{\mu}_2}{\partial V_{ij}}\right) \\
&= \sum_{k,l}\left(\frac{\partial\mathcal{L}}{\partial\boldsymbol{\mu}_2}^\top\right)_{kl}\left(\frac{\partial\boldsymbol{\mu}_2}{\partial S_{ij}^{-1}}\right)_{lk} \\
&= \left(\frac{\partial\mathcal{L}}{\partial\boldsymbol{\mu}_2}^\top\right)_{ji} = \left(\frac{\partial\mathcal{L}}{\partial\boldsymbol{\mu}_2}\right)_{ij}, \\
\Rightarrow \nabla_{\mathbf{V}}\mathcal{L} &= \nabla_{\boldsymbol{\mu}_2}\mathcal{L} \\
\Rightarrow \nabla_{(\mathbf{S})^{-1}}\mathcal{L}(\boldsymbol{\mu}) &= \left(\frac{\partial\mathcal{L}(\boldsymbol{\mu})}{\partial\boldsymbol{\mu}_2}\right)^\top \frac{\partial\boldsymbol{\mu}_2}{\partial(\mathbf{S})^{-1}} \\
&= \frac{1}{2}(\mathbf{H} - \gamma\mathbf{S}).
\end{aligned}
\tag{31}
$$

### A.2 Gradients in $(\mathbf{m}, \mathbf{C})$ Parametrization

Let us represent $\mathcal{L}(\boldsymbol{\mu}(\mathbf{C}))$ as $\mathcal{L}(\boldsymbol{\mu}_1, \boldsymbol{\mu}_2(\mathbf{C}))$ where $\boldsymbol{\mu}_1 = \mathbf{m}$ (independent of $\mathbf{C}$) and $\boldsymbol{\mu}_2(\mathbf{C}) = \mathbf{V} = \mathbf{CC}^\top$. Using the chain rule for composite matrix functions given in Equation 1884 in Dattorro (2015) combined with Equation 137 in Petersen & Pedersen (2012), we arrive at

$$
\begin{aligned}
\nabla_{\mathbf{C}}\mathcal{L}(\boldsymbol{\mu}(\mathbf{C})) &= \operatorname{tr}\left((\nabla_{\boldsymbol{\mu}_1}\mathcal{L}(\boldsymbol{\mu}))^\top (\nabla_{\mathbf{C}}\boldsymbol{\mu}_1(\mathbf{C}))\right) + \operatorname{tr}\left((\nabla_{\boldsymbol{\mu}_2}\mathcal{L}(\boldsymbol{\mu}))^\top \nabla_{\mathbf{C}}\boldsymbol{\mu}_2(\mathbf{C})\right) \\
&= \operatorname{tr}\left((\nabla_{\boldsymbol{\mu}_2}\mathcal{L}(\boldsymbol{\mu}))^\top \nabla_{\mathbf{C}}\boldsymbol{\mu}_2(\mathbf{C})\right),
\end{aligned}
$$

which leads to

$$
\frac{\partial\mathcal{L}(\boldsymbol{\mu}(\mathbf{C}))}{\partial C_{ij}} = \operatorname{tr}\left(\frac{\partial\mathcal{L}}{\partial\boldsymbol{\mu}_2}^\top \frac{\partial\boldsymbol{\mu}_2}{\partial C_{ij}}\right).
$$

Note that $\frac{\partial \boldsymbol{\mu}_2}{\partial \mathbf{C}}$ is a fourth-order tensor whose entries are

$$
\frac{\partial (\mathbf{C}\mathbf{C}^\top)_{kl}}{\partial C_{ij}} = \begin{cases} C_{lj} & k = i, l \neq i \\ 2C_{ij} & k = i, l = i \\ C_{kj} & k \neq i, l = i \\ 0 & \text{else.} \end{cases} \tag{32}
$$

where we used the expression $(\mathbf{C}\mathbf{C}^\top)_{kl} = \sum_n C_{kn} C_{nl}^\top = \sum_n C_{kn} C_{ln}$. Then

$$
\begin{aligned}
\frac{\partial \mathcal{L}(\boldsymbol{\mu}(\mathbf{C}))}{\partial C_{ij}} &= \operatorname{tr}\left( \frac{\partial \mathcal{L}}{\partial \boldsymbol{\mu}_2}^\top \frac{\partial \boldsymbol{\mu}_2}{\partial C_{ij}} \right) \\
&= \sum_{k,l} \left( \frac{\partial \mathcal{L}}{\partial \boldsymbol{\mu}_2}^\top \right)_{kl} \left( \frac{\partial \boldsymbol{\mu}_2}{\partial C_{ij}} \right)_{lk} \\
&= \sum_{k=i,l\neq i} \left( \frac{\partial \mathcal{L}}{\partial \boldsymbol{\mu}_2}^\top \right)_{kl} \left( \frac{\partial \boldsymbol{\mu}_2}{\partial C_{ij}} \right)_{lk} + \sum_{k=i,l=i} \left( \frac{\partial \mathcal{L}}{\partial \boldsymbol{\mu}_2}^\top \right)_{kl} \left( \frac{\partial \boldsymbol{\mu}_2}{\partial C_{ij}} \right)_{lk} + \sum_{k\neq i,l=i} \left( \frac{\partial \mathcal{L}}{\partial \boldsymbol{\mu}_2}^\top \right)_{kl} \left( \frac{\partial \boldsymbol{\mu}_2}{\partial C_{ij}} \right)_{lk} \\
&= \sum_{l\neq i} \left( \frac{\partial \mathcal{L}}{\partial \boldsymbol{\mu}_2}^\top \right)_{il} C_{lj} + 2 \left( \frac{\partial \mathcal{L}}{\partial \boldsymbol{\mu}_2}^\top \right)_{ii} C_{ij} + \sum_{k\neq i} \left( \frac{\partial \mathcal{L}}{\partial \boldsymbol{\mu}_2}^\top \right)_{ki} C_{kj} \\
&= \sum_{l} \left( \frac{\partial \mathcal{L}}{\partial \boldsymbol{\mu}_2}^\top \right)_{il} C_{lj} + \sum_{k} \left( \frac{\partial \mathcal{L}}{\partial \boldsymbol{\mu}_2}^\top \right)_{ki} C_{kj} \\
&= \left( \frac{\partial \mathcal{L}}{\partial \boldsymbol{\mu}_2}^\top \mathbf{C} \right)_{ij} + \left( \frac{\partial \mathcal{L}}{\partial \boldsymbol{\mu}_2} \mathbf{C} \right)_{ij}.
\end{aligned}
$$

Therefore, we conclude that

$$
\frac{\partial \mathcal{L}(\boldsymbol{\mu}_2(\mathbf{C}))}{\partial \mathbf{C}} = (\nabla_{\boldsymbol{\mu}_2} \mathcal{L})^\top \mathbf{C} + (\nabla_{\boldsymbol{\mu}_2} \mathcal{L}) \mathbf{C} = 2(\nabla_{\boldsymbol{\mu}_2} \mathcal{L}) \mathbf{C}, \tag{33}
$$

$$
\Rightarrow \frac{\partial \mathcal{L}(\boldsymbol{\mu}_2(\mathbf{C}))}{\partial \mathbf{C}} \overset{(29)}{=} \mathbf{H}\mathbf{C} - \gamma (\mathbf{C}^\top)^{-1} \tag{34}
$$

as $\nabla_{\boldsymbol{\mu}_2} \mathcal{L} = \nabla_{\mathbf{V}} \mathcal{L} = (\nabla_{\mathbf{V}} \mathcal{L})^\top$.

## B   SR-VN as an Approximation of VN (Detailed)

For $\gamma = 1$, the update of $\mathbf{S}_t$ from VN in Equation (7) can be rewritten in terms of $\mathbf{V}_t$ as,

$$
\begin{aligned}
\mathbf{V}_{t+1}^{-1} &= (1-\rho)\mathbf{V}_t^{-1} + \rho \mathbf{H}_t \\
&= (1-\rho)\mathbf{C}_t^{-\top}\mathbf{C}_t^{-1} + \rho \mathbf{H}_t \\
&= \mathbf{C}_t^{-\top}\left( (1-\rho)\mathbf{I} + \rho \mathbf{C}_t^\top \mathbf{H}_t \mathbf{C}_t \right) \mathbf{C}_t^{-1} \\
\Rightarrow \mathbf{V}_{t+1} &= \frac{\mathbf{C}_t}{(1-\rho)} \underbrace{\left( \mathbf{I} + \frac{\rho}{(1-\rho)} \mathbf{C}_t^\top \mathbf{H}_t \mathbf{C}_t \right)^{-1}}_{\text{Inverse}} \mathbf{C}_t^\top
\end{aligned}
$$

This above inverse can be approximated using the following truncated Neumann series for any symmetric matrix $\mathbf{A}$,

$$
(\mathbf{I} + \mathbf{A})^{-1} = \mathbf{I} - \mathbf{A} + \mathbf{A}^2 + \mathcal{O}(\|\mathbf{A}\|^3).
$$

to give

$$
\mathbf{V}_{t+1} \approx \frac{\mathbf{C}_t}{(1-\rho)} \left[ \mathbf{I} - \frac{\rho}{(1-\rho)} \mathbf{C}_t^\top \mathbf{H}_t \mathbf{C}_t + \left( \frac{\rho}{(1-\rho)} \right)^2 \mathbf{C}_t^\top \mathbf{H}_t \mathbf{C}_t \mathbf{C}_t^\top \mathbf{H}_t \mathbf{C}_t \right] \mathbf{C}_t^\top
$$

Let us define, $\mathbf{W}_t = \mathbf{C}_t^\top \mathbf{H}_t \mathbf{C}_t$, $\hat{\mathbf{W}}_t = \text{tril}[\mathbf{W}_t]$, and make use of the following identity: $\mathbf{W} = \hat{\mathbf{W}}_t + \hat{\mathbf{W}}_t^\top$ to arrive at

$$\mathbf{V}_{t+1} \approx \frac{\mathbf{C}_t}{(1-\rho)} \left[ \mathbf{I} - \frac{\rho}{(1-\rho)} \left[ \hat{\mathbf{W}}_t^\top + \hat{\mathbf{W}}_t \right] + \left( \frac{\rho}{(1-\rho)} \right)^2 \left[ \hat{\mathbf{W}}_t^\top + \hat{\mathbf{W}}_t \right]^2 \right] \mathbf{C}_t^\top$$

$$= \frac{\mathbf{C}_t}{(1-\rho)} \left[ \left( \mathbf{I} - \frac{\rho}{2(1-\rho)} \mathbf{W}_t \right)^2 + \frac{3}{4} \left( \frac{\rho}{(1-\rho)} \right)^2 \mathbf{W}_t^2 \right] \mathbf{C}_t^\top$$

$$\overset{(a)}{\approx} \mathbf{C}_t \left[ (1+\rho)\mathbf{I} - \rho\mathbf{W}_t \right] \mathbf{C}_t^\top + \mathcal{O}(\rho^2), \tag{35}$$

where $(a)$ comes from also expanding $(1-\rho)^{-1} = 1 + \rho + \rho^2 + \mathcal{O}(\rho^2)$, at all occurrences.

Now, let us write down the update for $\mathbf{V}_{t+1}$ using the updates from SR-VN (14) (writing $\alpha$ for the step-size for SR-VN here). We have

$$\mathbf{V}_{t+1} = \mathbf{C}_{t+1} \mathbf{C}_{t+1}^\top$$

$$= (\mathbf{C}_t - \alpha \mathbf{C}_t \, \text{tril}[\mathbf{C}_t^\top \mathbf{H}_t \mathbf{C}_t - \mathbf{I}])(\mathbf{C}_t - \alpha \mathbf{C}_t \, \text{tril}[\mathbf{C}_t^\top \mathbf{H}_t \mathbf{C}_t - \mathbf{I}])^\top$$

$$= \mathbf{C}_t \left[ \left( \left(1 + \frac{\alpha}{2}\right) \mathbf{I} - \alpha \, \text{tril}[\mathbf{C}_t^\top \mathbf{H}_t \mathbf{C}_t] \right) \left( \left(1 + \frac{\alpha}{2}\right) \mathbf{I} - \alpha \, \text{tril}[\mathbf{C}_t^\top \mathbf{H}_t \mathbf{C}_t] \right)^\top \right] \mathbf{C}_t^\top$$

$$= \mathbf{C}_t \left[ \left( \left(1 + \frac{\alpha}{2}\right) \mathbf{I} - \alpha \hat{\mathbf{W}}_t \right) \left( \left(1 + \frac{\alpha}{2}\right) \mathbf{I} - \alpha \hat{\mathbf{W}}_t \right)^\top \right] \mathbf{C}_t^\top$$

$$= \mathbf{C}_t \left[ \mathbf{I} + \alpha(\mathbf{I} - [\hat{\mathbf{W}}_t + \hat{\mathbf{W}}_t^\top]) + \alpha^2 \left( \frac{\mathbf{I} - 2[\hat{\mathbf{W}}_t + \hat{\mathbf{W}}_t^\top] + 4\hat{\mathbf{W}}_t \hat{\mathbf{W}}_t^\top}{4} \right) \right] \mathbf{C}_t^\top$$

$$= \mathbf{C}_t \left[ (1+\rho)\mathbf{I} - \rho\mathbf{W}_t \right] \mathbf{C}_t^\top + \mathcal{O}(\rho^2) \tag{36}$$

where $\alpha$ is chosen to be $\rho$. Comparing Equation (35) with Equation (36) shows that the two updates for $\mathbf{V}$ only match up to the first order. The presence of a quadratic difference in Equation (36) arises due to an alternative approximation of the inverse, highlighting the distinctions in the update methods and exposing the less-than-optimal nature of SR-VN.

## C  Analysis Proofs

Table 1: Vectorization Notation for a square matrix $\mathbf{A}$.

| | | |
|---:|:---:|:---|
| $\bar{\mathbf{A}}$ | $\overset{(\text{def})}{=}$ | lower triangular matrix derived from $\mathbf{A}$ by replacing all supra-diagonal elements by zero |
| $\text{diag}(\mathbf{A})$ | $\overset{(\text{def})}{=}$ | diagonal matrix derived from $\mathbf{A}$ by replacing all non-diagonal elements by zero |
| $\bar{\bar{\mathbf{A}}}$ | $\overset{(\text{def})}{=}$ | $\bar{\bar{\mathbf{A}}} - \bar{\mathbf{A}} - \text{diag}(\mathbf{A})/2$ |
| $\text{vec}(\mathbf{A})$ | $\overset{(\text{def})}{=}$ | vector obtained by stacking the columns of $\mathbf{A}$ in order from left to right |
| $\text{vech}(\mathbf{A})$ | $\overset{(\text{def})}{=}$ | vector obtained from $\text{vec}(\mathbf{A})$ by omitting supra-diagonal elements. |
| $\mathbf{K}$ | $\overset{(\text{def})}{=}$ | commutation matrix such that $\mathbf{K} \, \text{vec}(\mathbf{A}) = \text{vec}(\mathbf{A}^\top)$ |
| $\mathbf{L}$ | $\overset{(\text{def})}{=}$ | elimination matrix such that $\mathbf{L} \, \text{vec}(\mathbf{A}) = \text{vech}(\mathbf{A})$ |

### C.1 Proof of Lemma 1

*Proof.* In (Tan, 2025, Lemma 1), the inverse FIM in $(\mathbf{m}, \mathbf{C})$ parametrisation has the following form:

$$\mathbf{F}_{\boldsymbol{\tau}}^{-1}(\mathbf{m}, \mathbf{C}) = \begin{pmatrix} \mathbf{F}_{\mathbf{m}}^{-1} & 0 \\ 0 & \mathbf{F}_{\mathbf{C}}^{-1} \end{pmatrix} = \begin{pmatrix} \mathbf{V} & 0 \\ 0 & \mathbf{F}_{\mathbf{C}}^{-1} \end{pmatrix}, \tag{37}$$

where, given $\mathbf{N} = (\mathbf{K} + \mathrm{I}_{d^2})/2$,

$$\mathbf{F}_{\mathbf{C}}^{-1} = \frac{1}{2}\mathbf{L}(\mathrm{I}_d \otimes \mathbf{C})\mathbf{L}^\top(\mathbf{L}\mathbf{N}\mathbf{L}^\top)^{-1}\mathbf{L}(\mathrm{I}_d \otimes \mathbf{C}^\top)\mathbf{L}^\top. \tag{38}$$

Let us work first to prove the LHS of the claim. From part 2 of Assumption 4, we first see that $\mathbf{F}_{\mathbf{m}}^{-1} = \mathbf{V} \succcurlyeq \lambda_{\min}\mathrm{I}$. For $\mathbf{F}_{\mathbf{C}}^{-1}$, we proceed as follows:

$$\left\|\mathbf{F}_{\mathbf{C}}^{-1}\right\|_2 \geq \frac{1}{\|\mathbf{F}_{\mathbf{C}}\|_2} \stackrel{(Tan,\ 2025,\ \text{Lemma 1(i)})}{=} \frac{1}{\left\|2\mathbf{L}(\mathrm{I} \otimes \mathbf{C}^{-\top})\mathbf{N}(\mathrm{I} \otimes \mathbf{C}^{-1})\mathbf{L}^\top\right\|_2}.$$

Now, notice that $\|\mathbf{N}\|_2 \leq \frac{1}{2}(\|\mathbf{K}\|_2 + \|\mathrm{I}_{d^2}\|_2) \leq 1$ since $\|\mathbf{K}\|_2 \leq 1$, while $\mathbf{K}$ being a permutation matrix. Since $\mathbf{L}$ is also form of a permutation matrix (with additional zeros at appropriate places), we have that $\|\mathbf{K}\|_2 \leq 1$. Combining these facts with the norm inequality, we obtain

$$\left\|\mathbf{F}_{\mathbf{C}}^{-1}\right\|_2 \geq \frac{1}{\left\|2\mathbf{L}(\mathrm{I} \otimes \mathbf{C}^{-\top})\mathbf{N}(\mathrm{I} \otimes \mathbf{C}^{-1})\mathbf{L}^\top\right\|_2} \geq \frac{1}{2\|\mathbf{L}\|_2^2 \|\mathbf{N}\|_2 \|\mathrm{I} \otimes \mathbf{C}^{-\top}\|_2 \|\mathrm{I} \otimes \mathbf{C}^{-1}\|_2}$$

$$\geq \frac{1}{2\|\mathrm{I} \otimes \mathbf{C}^{-\top}\|_2 \|\mathrm{I} \otimes \mathbf{C}^{-1}\|_2}$$

$$\stackrel{(a)}{\geq} \frac{1}{2\|\mathbf{C}^{-1}\|_2^2}$$

$$\geq \frac{\lambda_{\min}^2}{2} \qquad \left(\because \|\mathbf{C}^{-1}\|_2^2 \geq \|\mathbf{V}^{-1}\|_2 = \frac{1}{\lambda_{\min}}\right)$$

where $(a)$ comes from using $\|\mathbf{A} \otimes \mathbf{B}\|_2 \leq \|\mathbf{A}\|_2 \|\mathbf{B}\|_2$, $\forall \mathbf{A}, \mathbf{B} \in \mathbb{R}^{d \times d}$, and $\|\mathbf{B}^\top\|_2 = \|\mathbf{B}\|_2$.

Now let us prove the RHS of the claim. From Remark 1, we have $\mathbf{F}_{\mathbf{m}}^{-1} = \mathbf{V} \preccurlyeq \xi_u^2\mathrm{I}$. For $\mathbf{F}_{\mathbf{C}}^{-1}$, we use the norm inequality as follows:

$$\left\|\mathbf{F}_{\mathbf{C}}^{-1}\right\|_2 \leq \frac{1}{2}\|\mathbf{L}\|_2^4 \left\|(\mathbf{L}\mathbf{N}\mathbf{L}^\top)^{-1}\right\|_2 \|\mathrm{I}_d \otimes \mathbf{C}\|_2 \|\mathrm{I}_d \otimes \mathbf{C}^\top\|_2$$

$$\leq \frac{1}{2}\left\|(\mathbf{L}\mathbf{N}\mathbf{L}^\top)^{-1}\right\|_2 \|\mathbf{C}\|_2^2$$

$$\stackrel{\text{Assumption 4}}{\leq} \frac{\xi_u^2}{2}\left\|(\mathbf{L}\mathbf{N}\mathbf{L}^\top)^{-1}\right\|_2 \tag{39}$$

Let us see how the operator $\mathbf{L}\mathbf{N}\mathbf{L}^\top : \mathbb{R}^{d(d+1)/2} \to \mathbb{R}^{d(d+1)/2}$ acts on $\mathrm{vech}(\mathbf{A}) \in \mathbb{R}^{d(d+1)/2}$. By definition, we have

$$(\mathbf{L}\mathbf{N}\mathbf{L}^\top)\,\mathrm{vech}(\mathbf{A}) = \left(\mathbf{L}\,\frac{\mathbf{K} + \mathrm{I}_{d^2}}{2}\,\mathbf{L}^\top\right)\mathrm{vech}(\mathbf{A})$$

$$= \frac{1}{2}((\mathbf{L}\mathbf{K}\mathbf{L}^\top)\,\mathrm{vech}(\mathbf{A}) + \underbrace{(\mathbf{L}\mathbf{L}^\top)}_{\mathrm{I}_{d(d+1)/2}}\,\mathrm{vech}(\mathbf{A}))$$

$$= \frac{1}{2}(\mathbf{L}\mathbf{K}\,\mathrm{vec}(\bar{\mathbf{A}}) + \mathrm{vech}(\mathbf{A})), \qquad (\because \mathbf{L}^\top\,\mathrm{vech}(\mathbf{A}) = \mathrm{vec}(\bar{\mathbf{A}}))$$

$$= \frac{1}{2}(\mathbf{L}\,\mathrm{vec}(\bar{\mathbf{A}}^\top) + \mathrm{vech}(\mathbf{A})), \qquad (\because \mathbf{K}\,\mathrm{vec}(\bar{\mathbf{A}}) = \mathrm{vec}(\bar{\mathbf{A}}^\top))$$

$$= \frac{1}{2}(\mathrm{vech}(\bar{\mathbf{A}}^\top) + \mathrm{vech}(\mathbf{A}))), \qquad (\because \mathbf{L}\,\mathrm{vec}(\bar{\mathbf{A}}^\top) = \mathrm{vech}(\bar{\mathbf{A}}^\top)), \tag{40}$$

where Equation (40) implies that the operator $\mathbf{LNL}^\top$ is equivalent to halving the sub-diagonal entries while keeping the diagonal of the matrix $\mathbf{A}$ intact. Therefore, $\mathbf{LNL}^\top$ has eigenvalues between $1/2$ and $1$, which implies that its inverse has eigenvalues bounded between 1 and 2, leading to $\left\|(\mathbf{LNL}^\top)^{-1}\right\|_2 \leq 2$. Thus, after plugging in Equation (39) this bound, we have

$$\left\|\mathbf{F}_{\mathbf{C}}^{-1}\right\|_2 \leq \xi_u^2, \tag{41}$$

allowing us to complete the proof of the main claim. □

## C.2   Proof of Lemma 2

*Proof.* Notice that Equation (20) is equivalent to

$$\|\nabla_{\boldsymbol{\tau}}\mathcal{L}(\boldsymbol{\tau})\|_{\mathbf{F}_{\boldsymbol{\tau}}^{-1}}^2 \geq 2\mu\left(\mathcal{L}(\boldsymbol{\tau}) - \mathcal{L}(\boldsymbol{\tau}_*)\right), \tag{42}$$

because $\mathrm{KL}\left[q_{\boldsymbol{\tau}}(\boldsymbol{\theta}) \,\|\, p(\boldsymbol{\theta})\right]$ can be written as

$$\mathrm{KL}\left[q_{\boldsymbol{\tau}}(\boldsymbol{\theta}) \,\|\, p(\boldsymbol{\theta})\right] = -1/2 \log(\det \mathbf{V}) + \mathbb{E}_{\boldsymbol{\theta} \sim \mathcal{N}(\mathbf{m}, \mathbf{V})}[\bar{\ell}(\boldsymbol{\theta})] + \mathrm{const.} \overset{(2)}{=} \mathcal{L}(\boldsymbol{\tau}) + \mathrm{const.}$$

Then simply note that by Lemma 1, we obtain that

$$\|\nabla_{\boldsymbol{\tau}}\mathcal{L}\|_{\mathbf{F}_{\boldsymbol{\tau}}^{-1}}^2 \geq \lambda_{\min}^g \|\nabla_{\boldsymbol{\tau}}\mathcal{L}\|^2 \overset{(16)}{\geq} 2\delta\lambda_{\min}^g\left(\mathcal{L}(\mathbf{m}, \mathbf{C}) - \mathcal{L}(\mathbf{m}_*, \mathbf{C}_*)\right)$$

where now $\mu = \delta\lambda_{\min}$ is indeed the desired the PL constant. □

## C.3   Proof of Theorem 1

*Proof.* For conciseness, we will refer $\mathrm{KL}\left[q_{\boldsymbol{\tau}_t}(\boldsymbol{\theta}) \,\|\, p(\boldsymbol{\theta} \mid \mathcal{D})\right] = \mathrm{KL}_t$ throughout the proof. Let us define the Lyapunov function $\mathcal{E}(t) = e^{2\mu t}(\mathrm{KL}_t - \mathrm{KL}_*)$, then

$$\dot{\mathcal{E}}_t = 2\mu e^{2\mu t}(\mathrm{KL}_t - \mathrm{KL}_*) + e^{2\mu t}\langle\nabla_{\boldsymbol{\tau}}\mathrm{KL}_t, \dot{\boldsymbol{\tau}}\rangle \overset{(8)}{=} e^{2\mu t}(2\mu(\mathrm{KL}_t - \mathrm{KL}_*) - \|\nabla_{\boldsymbol{\tau}}\mathrm{KL}_t\|_{\mathbf{F}_{\boldsymbol{\tau}}^{-1}}^2) \overset{\mathrm{Lemma\ 2}}{\leq} 0.$$

The energy functional $\mathcal{E}(t)$ is therefore monotonically decreasing, and hence

$$\mathcal{E}(t) = e^{2\mu t}(\mathrm{KL}_t - \mathrm{KL}_*) \leq \mathcal{E}(0) = \mathrm{KL}_0 - \mathrm{KL}_* \tag{43}$$

□

## C.4   Proof of Theorem 2

*Proof.* From Assumption 3, we start by writing the following Taylor expansion for $\mathcal{L}(\mathbf{m}, \mathbf{C})$ when both $\mathbf{m}$ and $\mathbf{C}$ change.

$$\mathcal{L}(\mathbf{m}_{t+1}, \mathbf{C}_{t+1}) \leq \mathcal{L}(\mathbf{m}_t, \mathbf{C}_t) + \nabla_{\mathbf{m}}\mathcal{L}(\mathbf{m}_t, \mathbf{C}_t)^\top(\mathbf{m}_{t+1} - \mathbf{m}_t) + \mathrm{tr}((\nabla_{\mathbf{C}}\mathcal{L}(\mathbf{m}_t, \mathbf{C}_t))^\top(\mathbf{C}_{t+1} - \mathbf{C}_t))$$
$$+ \frac{M}{2}\|\mathbf{m}_{t+1} - \mathbf{m}_t\|_2^2 + \frac{M}{2}\|\mathbf{C}_{t+1} - \mathbf{C}_t\|_F^2. \tag{44}$$

By plugging in values from Equation (14) in the joint Taylor expansion above, we arrive at:

$$\mathcal{L}(\mathbf{m}_{t+1}, \mathbf{C}_{t+1}) - \mathcal{L}(\mathbf{m}_t, \mathbf{C}_t)$$
$$\leq \underbrace{-(\nabla_{\mathbf{m}}\mathcal{L}(\mathbf{m}_t, \mathbf{C}_t)^\top(\rho\mathbf{C}_t\mathbf{C}_t^\top\nabla_{\mathbf{m}}\mathcal{L}(\mathbf{m}_t, \mathbf{C}_t)) + \frac{M\rho^2}{2}\left\|\mathbf{C}_t\mathbf{C}_t^\top\nabla_{\mathbf{m}}\mathcal{L}(\mathbf{m}_t, \mathbf{C}_t)\right\|^2}_{A}$$
$$\underbrace{-\rho\,\mathrm{tr}\left((\nabla_{\mathbf{C}}\mathcal{L}(\mathbf{m}_t, \mathbf{C}_t))^\top\mathbf{C}_t\bar{\bar{\mathbf{H}}}_t\right) + \frac{M\rho^2}{2}\|\mathbf{C}_t\bar{\bar{\mathbf{H}}}_t\|_F^2}_{B}, \tag{45}$$

where $\bar{\bar{\mathbf{H}}}_t = \mathrm{tril}[\mathbf{C}_t^\top \nabla_{\mathbf{C}}\mathcal{L}(\mathbf{m}_t, \mathbf{C}_t)]$. Using the following facts

$$-\nabla_{\mathbf{m}}\mathcal{L}(\mathbf{m}_t, \mathbf{C}_t)^\top \mathbf{C}_t \mathbf{C}_t^\top \nabla_{\mathbf{m}}\mathcal{L}(\mathbf{m}_t, \mathbf{C}_t) \le -\lambda_{\min} \|\nabla_{\mathbf{m}}\mathcal{L}(\mathbf{m}_t, \mathbf{C}_t)\|^2$$

$$\left\|\mathbf{C}_t \mathbf{C}_t^\top \nabla_{\mathbf{m}}\mathcal{L}(\mathbf{m}_t, \mathbf{C}_t)\right\| \le \left\|\mathbf{C}_t \mathbf{C}_t^\top\right\| \|\nabla_{\mathbf{m}}\mathcal{L}(\mathbf{m}_t, \mathbf{C}_t)\| \overset{\text{Remark 1}}{\le} \xi_u^2 \|\nabla_{\mathbf{m}}\mathcal{L}(\mathbf{m}_t, \mathbf{C}_t)\|$$

and applying the lower for bound $\lambda_{\min}$ assumed in Assumption 4 to term $A$, we arrive at

$$A \le -\lambda_{\min}\rho \|\nabla_{\mathbf{m}}\mathcal{L}(\mathbf{m}_t, \mathbf{C}_t)\|^2 + \frac{M\rho^2 \xi_u^4}{2} \|\nabla_{\mathbf{m}}\mathcal{L}(\mathbf{m}_t, \mathbf{C}_t)\|^2,$$

which then gives us

$$A \le \omega_m \|\nabla_{\mathbf{m}}\mathcal{L}(\mathbf{m}_t, \mathbf{C}_t)\|^2, \tag{46}$$

where

$$\omega_m = \frac{M\rho^2 \xi_u^4}{2} - \lambda_{\min}\rho.$$

Now for term $B$, let us first look at the following term by repeatedly applying the matrix norm inequality $(\|\mathbf{AB}\|_F \le \|\mathbf{A}\|_F \|\mathbf{B}\|_F)$ to obtain

$$\|\mathbf{C}_t \bar{\bar{\mathbf{H}}}_t\|_F^2 \le \|\mathbf{C}_t\|_F^2 \left\|\bar{\bar{\mathbf{H}}}_t\right\|_F^2 \overset{(a)}{\le} \|\mathbf{C}_t\|_F^2 \left(2\|\bar{\mathbf{H}}\|_F^2 + \frac{1}{2}\|\mathrm{diag}(\mathbf{H})\|_F^2\right) \overset{(b)}{\le} \|\mathbf{C}_t\|_F^2 \left(2\|\mathbf{H}\|_F^2 + \frac{1}{2}\|\mathrm{diag}(\mathbf{H})\|_F^2\right)$$

$$\le \|\mathbf{C}_t\|_F^2 \left(2\|\mathbf{H}\|_F^2 + \frac{1}{2}\|\mathbf{H}\|_F^2\right) = \frac{5}{2}\|\mathbf{C}_t\|_F^2 \|\mathbf{H}\|_F^2$$

$$= \frac{5}{2}\|\mathbf{C}_t\|_F^2 \left\|\mathbf{C}^\top \bar{\mathbf{K}}\right\|_F^2 \le \frac{5}{2}\|\mathbf{C}_t\|_F^4 \left\|\bar{\mathbf{K}}\right\|_F^2 \overset{(c)}{\le} \frac{5}{2}\|\mathbf{C}_t\|_F^4 \|\mathbf{K}\|_F^2$$

$$= \frac{5}{2}\|\mathbf{C}_t\|_F^4 \|\nabla_{\mathbf{C}}\mathcal{L}(\mathbf{m}_t, \mathbf{C}_t)\|_F^2,$$

where $(a)$ comes from using the fact that $\forall \mathbf{A}, \mathbf{B} \in \mathbb{R}^{D \times D}$, we have $\|\mathbf{A} - \mathbf{B}\|_F^2 \le 2\|\mathbf{A}\|_F^2 + 2\|\mathbf{B}\|_F^2$, and choosing $\mathbf{A} = \bar{\mathbf{H}}$ and $\mathbf{B} = \mathrm{diag}(\mathbf{H})/2$. In $(b)$ and $(c)$ we have used $\|\bar{\mathbf{A}}\|_F \le \|\mathbf{A}\|_F$, as $\bar{\mathbf{A}}$ is a lower triangular form obtained by sending all supra-diagonal elements of $\mathbf{A}$ to zero. For the first term in $B$, we have

$$\mathrm{tr}\left((\nabla_{\mathbf{C}}\mathcal{L}(\mathbf{m}_t, \mathbf{C}_t))^\top \mathbf{C}_t \bar{\bar{\mathbf{H}}}_t\right) \le \|\nabla_{\mathbf{C}}\mathcal{L}(\mathbf{m}_t, \mathbf{C}_t)\|_F \|\mathbf{C}_t \bar{\bar{\mathbf{H}}}_t\|_F \qquad (\because \mathrm{tr}(\mathbf{AB}) \le \|\mathbf{A}\|_F \|\mathbf{B}\|_F)$$

$$\le \sqrt{\frac{5}{2}} \|\mathbf{C}_t\|_F^2 \|\nabla_{\mathbf{C}}\mathcal{L}(\mathbf{m}_t, \mathbf{C}_t)\|_F^2$$

This leads us to finally obtain

$$B \le -\sqrt{\frac{5}{2}}\rho \|\mathbf{C}_t\|_F^2 \|\nabla_{\mathbf{C}}\mathcal{L}(\mathbf{m}_t, \mathbf{C}_t)\|_F^2 + \frac{5M\rho^2}{4}\|\mathbf{C}_t\|_F^4 \|\nabla_{\mathbf{C}}\mathcal{L}(\mathbf{m}_t, \mathbf{C}_t)\|_F^2$$

Now using the bounds in Assumption 4, we arrive at

$$B \le \omega_C \|\nabla_{\mathbf{C}}\mathcal{L}(\mathbf{m}_t, \mathbf{C}_t)\|_F^2 \tag{47}$$

where

$$\omega_C = \frac{5\rho^2 \xi_u^4 M}{4} - \sqrt{\frac{5}{2}}\rho \xi_l^2.$$

Now plugging Equation (46) and Equation (47) in Equation (45) to get

$$\Delta^{t+1} - \Delta^t \le \omega_m \|\nabla_{\mathbf{m}}\mathcal{L}(\mathbf{m}_t, \mathbf{C}_t)\|^2 + \omega_C \|\nabla_{\mathbf{C}}\mathcal{L}(\mathbf{m}_t, \mathbf{C}_t)\|_F^2$$

$$\Rightarrow \quad \Delta^{t+1} - \Delta^t \le -\eta \left\|\nabla_{(\mathbf{m}, \mathbf{C})}\mathcal{L}(\mathbf{m}_t, \mathbf{C}_t)\right\|^2, \tag{48}$$

where $\eta = \min\{-\omega_m, -\omega_C\}$ and we require that $\eta > 0$. Now from Assumption 2, by joint $\delta$-strongly convexity of $\mathcal{L}$ in $(\mathbf{m}, \mathbf{C})$, we have

$$\mathcal{L}(\mathbf{m}_t, \mathbf{C}_t) - \mathcal{L}(\mathbf{m}^*, \mathbf{C}^*) \leq \frac{1}{2\delta} \left\| \nabla_{(\mathbf{m}, \mathbf{C})} \mathcal{L}(\mathbf{m}_t, \mathbf{C}_t) \right\|^2$$

$$\Rightarrow \Delta^t \leq \frac{1}{2\delta} \left\| \nabla_{(\mathbf{m}, \mathbf{C})} \mathcal{L}(\mathbf{m}_t, \mathbf{C}_t) \right\|^2$$

$$\Rightarrow \quad \Delta^t \overset{(48)}{\leq} \frac{1}{2\eta\delta} (\Delta^t - \Delta^{t+1})$$

$$\Rightarrow \quad \Delta^{t+1} \leq (1 - 2\eta\delta) \Delta^t \tag{49}$$

from which we can conclude that we can indeed obtain a descent for $\mathcal{L}$ in $(\mathbf{m}, \mathbf{C})$ if we require

$$0 < 1 - 2\eta\delta < 1 \Rightarrow 0 < \eta < \frac{1}{2\delta} \tag{50}$$

This would further lead us to the global convergence as

$$\Delta^{t+1} \leq (1 - 2\eta\delta)^{t+1} \Delta^0. \tag{51}$$

$\square$

## C.5 Permissible Step Sizes for SR-VN

Theorem 2 requires the condition $0 < \eta < \frac{1}{2\delta}$ to hold. This might prompt the reader to inquire whether a specific step size $\eta$ can be determined to ensure the fulfillment of this condition. We answer this question below in the context of a constant step size that we also employ in our experimental work.

**Constant step size.** Let $\lambda_{\max} = \xi_u^2$ be the upper bound for the largest eigenvalue of $\mathbf{V}_t$, as derived in Remark 1, then Equation (22) becomes

$$\eta = \min \left\{ \lambda_{\min}\rho - \frac{M\rho^2\xi_u^4}{2}, \quad \sqrt{\frac{5}{2}}\rho\xi_l^2 - \frac{5\rho^2\xi_u^4 M}{4} \right\}.$$

Note that both arguments inside the min above are quadratics in $\rho$. Let us choose $\rho$ such that

$$\rho \leq \max\{\rho_1, \rho_2\}, \tag{52}$$

where

$$\rho_1 = \frac{\lambda_{\min}}{\lambda_{\max}^2 M} \quad \text{and} \quad \rho_2 = \sqrt{\frac{2}{5}} \frac{\xi_l^2}{M\xi_u^4}$$

are the extremal points of the quadratic equations in the first and second terms inside the min, respectively. Choosing $\rho = \rho_1$ for the first argument and $\rho = \rho_2$ for the second argument yields

$$\eta \leq \min \left\{ \frac{\lambda_{\min}^2}{2\lambda_{\max}^2 M}, \frac{\xi_l^4}{2M\xi_u^4} \right\}. \tag{53}$$

In order to ensure that the rate of convergence is contractive, we require $(1 - 2\eta\delta) < 1$, that is $0 < 2\eta\delta < 1$. With the choice above, $\eta$ is obviously positive, so let us focus on $2\eta\delta < 1 \implies \eta < \frac{1}{2\delta}$. From the first quantity in the min appearing in Equation (53), we need

$$\frac{\lambda_{\min}^2}{2\lambda_{\max}^2 M} < \frac{1}{2\delta} \implies \frac{\lambda_{\min}^2}{\lambda_{\max}^2} \cdot \frac{\delta}{M} < 1. \tag{54}$$

The quantity $\frac{\lambda_{\min}}{\lambda_{\max}}$ is a lower bound on the inverse condition number of the preconditioning matrix $\mathbf{V}_t$ and is thus upper bounded by 1. Since $M$ is the smoothness constant of $\mathcal{L}(\mathbf{m}, \mathbf{C})$, then the quantity $\frac{\delta}{M}$ is upper bounded by the inverse condition number of the Hessian of $\mathcal{L}(\mathbf{m}, \mathbf{C})$ which is itself upper bounded by 1.

This means that Equation (54) is always satisfied with the proposed choice of step size. The same argument applies to the second quantity in the min appearing in Equation (53).

**Intuition:** We here give an intuitive explanation for the choice of the step size $\rho$. By Theorem 2, we see that the speed of convergence is exponential with respect to time $t$. The contraction factor $C_t := (1 - 2\eta\delta)^{t+1}$ directly depends on the choice of step size through $\eta$ defined in Equation (22). From the latter equation, one can observe that larger step sizes can be used if the objective function is smoother (which is controlled by $M$). In turn, larger step sizes allow for faster convergence as the algorithm takes larger steps towards the minimum, which means that $(1 - 2\eta\delta)$ is smaller and therefore $C_t$ contracts at a faster rate.

**Complexity Calculation.** To compute the complexity, we require $\mathcal{L}(\mathbf{m}_t, \mathbf{C}_t) - \mathcal{L}(\mathbf{m}_*, \mathbf{C}_*) \leq \epsilon$, meaning from Equation (21) we must have

$$(1 - 2\eta\delta)^t (\mathcal{L}(\mathbf{m}_0, \mathbf{C}_0) - \mathcal{L}(\mathbf{m}_*, \mathbf{C}_*)) \leq \epsilon \implies t \leq \frac{\log(\epsilon/(\mathcal{L}(\mathbf{m}_0, \mathbf{C}_0) - \mathcal{L}(\mathbf{m}_*, \mathbf{C}_*)))}{\log(1 - 2\eta\delta)}.$$

Now using Equation (53), we arrive at

$$t \geq \min\left\{ \frac{\log(\epsilon/(\mathcal{L}(\mathbf{m}_0, \mathbf{C}_0) - \mathcal{L}(\mathbf{m}_*, \mathbf{C}_*)))}{\log(1 - \frac{\delta}{M}\frac{\lambda_{\min}^2}{\lambda_{\max}^2})}, \frac{\log(\epsilon/(\mathcal{L}(\mathbf{m}_0, \mathbf{C}_0) - \mathcal{L}(\mathbf{m}_*, \mathbf{C}_*)))}{\log(1 - \frac{\delta}{M}\frac{\xi_l^4}{\xi_u^4})} \right\}, \tag{55}$$

where the switching of inequality happens because the $\log(\cdot)$ term in the denominator is negative for both terms inside min.

# D  Biased SR-VN Convergence

**Definition 1** (Biased Gradient Oracle). *A map* $\mathbf{g} : \mathbb{R}^{d+d\times d} \times \mathcal{D} \to \mathbb{R}^d$ *s.t.*

$$\hat{\mathbf{g}}(\boldsymbol{\tau}) = \mathbf{g} + \mathbf{b}_g(\boldsymbol{\tau}). \tag{56}$$

*for a bias* $b_g : \mathbb{R}^{d+d\times d} \to \mathbb{R}^d$

**Definition 2** (Biased Hessian Oracle). *A map* $\mathbf{H} : \mathbb{R}^{d+d\times d} \times \mathcal{D} \to \mathbb{R}^{d\times d}$ *s.t.*

$$\hat{\mathbf{H}}(\boldsymbol{\tau}) = \mathbf{H} + \mathbf{b}_H(\boldsymbol{\tau}). \tag{57}$$

*for a bias* $b_H : \mathbb{R}^{d+d\times d} \to \mathbb{R}^{d\times d}$

We assume that the bias is bounded:

**Assumption 5.** *(Bounded biases) There exists constants* $0 \leq m_g, m_H < 1$, *and* $\zeta_m^2, \zeta_H^2 \geq 0$ *such that*

1. $\|\mathbf{b}_g(\boldsymbol{\tau})\|^2 \leq m_g \|\nabla_{\mathbf{m}}\mathcal{L}(\mathbf{m}_t, \mathbf{C}_t)\|^2 + \zeta_g^2, \quad \forall \boldsymbol{\tau} \in \mathbb{R}^{d+d\times d}.$

2. $\|\mathbf{b}_H(\boldsymbol{\tau})\mathbf{C}\|_F^2 \leq m_H \|\nabla_{\mathbf{C}}\mathcal{L}(\mathbf{m}_t, \mathbf{C}_t)\|_F + \zeta_H^2, \quad \forall \boldsymbol{\tau} \in \mathbb{R}^{d+d\times d}.$

With the prerequisites established, we now present our theorem demonstrating the convergence of the biased SR-VN update. This bias may arise from estimating the expectation using piecewise bounds presented in Marlin et al. (2011).

**Theorem 3** (With Bias Convergence). *Given Assumptions 1–5 and considering the limit points* $\mathbf{m}_*$ *and* $\mathbf{C}_*$ *in Equation* (19), *the updates provided by Algorithm 1 satisfy*

$$\mathcal{L}(\mathbf{m}_{t+1}, \mathbf{C}_{t+1}) - \mathcal{L}(\mathbf{m}_*, \mathbf{C}_*) \leq (1 - 2\eta\delta)^{t+1}(\mathcal{L}(\mathbf{m}_0, \mathbf{C}_0) - \mathcal{L}(\mathbf{m}_*, \mathbf{C}_*)) + \omega_m \zeta_g^2 + \omega_C \zeta_H^2,$$

*Proof.* Following similar steps, we can arrive at the following versions of Equation (46) and Equation (47)

$$A \leq \omega_m \left\|\hat{\mathbf{H}}_t\right\|^2, \qquad B \leq \omega_C \|\nabla_{\mathbf{C}}\mathcal{L}(\mathbf{m}_t, \mathbf{C}_t)\|_F^2, \tag{58}$$

where $\nabla_{\mathbf{C}}\mathcal{L}(\mathbf{m}_t, \mathbf{C}_t) = (\hat{\mathbf{H}}_t - \gamma(\mathbf{C}_t\mathbf{C}_t^\top)^{-1})\mathbf{C}_t$. Now using Assumption 5, one can get

$$A \leq \omega_m \left\|\hat{\mathbf{H}}_t\right\|^2 = \omega_m \|\mathbf{H}_t + \mathbf{b}_g(\boldsymbol{\tau}_t)\|^2 \leq \omega_m \left( \|\mathbf{H}_t\|^2 + \|\mathbf{b}_g(\boldsymbol{\tau}_t)\|^2 \right) \leq \omega_m \left( (1 + m_g) \|\mathbf{H}_t\|^2 + \zeta_g^2 \right). \quad (59)$$

Similarly,

$$\begin{aligned}
B &\leq \omega_C \|\nabla_{\mathbf{C}}\mathcal{L}(\mathbf{m}_t, \mathbf{C}_t)\|_F^2 = \omega_C \left\|\left(\mathbf{H}_t + \mathbf{b}_H(\boldsymbol{\tau}_t) - \gamma(\mathbf{C}_t\mathbf{C}_t^\top)^{-1}\right)\mathbf{C}_t\right\|_F^2 \\
&\leq \omega_C \left\|\left(\mathbf{H}_t - \gamma(\mathbf{C}_t\mathbf{C}_t^\top)^{-1}\right)\mathbf{C}_t\right\|_F^2 + \omega_C \|\mathbf{b}_H(\boldsymbol{\tau}_t)\mathbf{C}_t\|_F^2 \\
&= \omega_C \left( \|\nabla_{\mathbf{C}}\mathcal{L}(\mathbf{m}_t, \mathbf{C}_t)\|_F^2 + \|\mathbf{b}_H(\boldsymbol{\tau}_t)\mathbf{C}_t\|_F^2 \right) \\
&\leq \omega_C \left( (1 + m_H) \|\nabla_{\mathbf{C}}\mathcal{L}(\mathbf{m}_t, \mathbf{C}_t)\|^2 + \zeta_H^2 \right)
\end{aligned} \quad (60)$$

Now plugging Equation (59) and Equation (60) in Equation (45) to get

$$\begin{aligned}
&\Delta^{t+1} - \Delta^t \\
&\leq \omega_m(1 + m_g) \|\nabla_{\mathbf{m}}\mathcal{L}(\mathbf{m}_t, \mathbf{C}_t)\|^2 + \omega_C(1 + m_H) \|\nabla_{\mathbf{C}}\mathcal{L}(\mathbf{m}_t, \mathbf{C}_t)\|_F^2 + \omega_m\zeta_g^2 + \omega_C\zeta_H^2 \quad (61) \\
\Rightarrow \quad &\Delta^{t+1} - \Delta^t \leq -\eta \left\|\nabla_{(\mathbf{m},\mathbf{C})}\mathcal{L}(\mathbf{m}_t, \mathbf{C}_t)\right\|^2 + \omega_m\zeta_g^2 + \omega_C\zeta_H^2, \quad (62)
\end{aligned}$$

where $\eta = \min\{-\omega_m(1 + m_g), -\omega_C(1 + m_H)\}$ and we require that $\eta > 0$. Now from Assumption 2, by joint $\delta$-strongly convexity of $\mathcal{L}$ in $(\mathbf{m}, \mathbf{C})$, we have

$$\begin{aligned}
\mathcal{L}(\mathbf{m}_t, \mathbf{C}_t) - \mathcal{L}(\mathbf{m}^*, \mathbf{C}^*) &\leq \frac{1}{2\delta} \left\|\nabla_{(\mathbf{m},\mathbf{C})}\mathcal{L}(\mathbf{m}_t, \mathbf{C}_t)\right\|^2 \\
\Rightarrow \Delta^t &\leq \frac{1}{2\delta} \left\|\nabla_{(\mathbf{m},\mathbf{C})}\mathcal{L}(\mathbf{m}_t, \mathbf{C}_t)\right\|^2 \\
\Rightarrow \quad \Delta^t &\overset{(62)}{\leq} \frac{1}{2\eta\delta}(\Delta^t - \Delta^{t+1}) + \frac{1}{2\eta\delta}(\omega_m\zeta_g^2 + \omega_C\zeta_H^2) \\
\Rightarrow \quad \Delta^{t+1} &\leq (1 - 2\eta\delta)\Delta^t + \omega_m\zeta_g^2 + \omega_C\zeta_H^2 \quad (63)
\end{aligned}$$

$\square$

*Remark* 3. It is worth noting that the piecewise method for computing expectations, given in Marlin et al. (2011), involves computing a 1-D integral which one can compute below numerical float precision. In other words, the bias with piecewise bounds can be made arbitrarily small by increasing the number of pieces. The bias is therefore very small, and hence the terms $\zeta_g^2$ and $\zeta_H^2$ are negligible which shows that SR-VN converges as long as this holds.

## E   Piecewise versus stochastic implementation

In this section, we make a comparison between the piecewise and the stochastic implementation of the expectation in ELBO, see Figure 4. This plot highlights why the piecewise implementation is a more accurate and faster representation of the deterministic algorithm we analyzed, and thus the preferred choice for the experiments carried out in the paper.

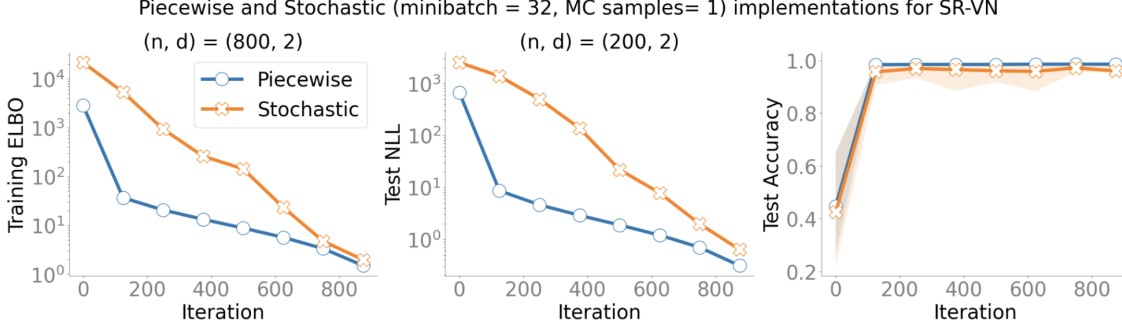

Figure 4: The provided comparison highlights the differences between the piecewise implementation and the (minibatch + MC sampling) stochastic implementation for SR-VN. The results clearly demonstrate that the stochastic approach is much slower compared to the piecewise method. Additionally, while the stochastic implementation converges to a slightly higher ELBO, the piecewise implementation reaches a more accurate solution more rapidly. These findings are based on iterates averaged over 5 different initializations. The dataset taken is the same as displayed in Figure 5, taken with different $n$'s.

## F   Experiments Detail

For all experiments, we first use grid search to tune model hyper-parameters, where the search is performed in a specific range of values. The resultant values were then fixed during our experiments. The statistics of the datasets and the model hyper-parameters used are given in Table 2. In practice, implementing deterministic NGD is often feasible by employing piecewise analytical linear and quadratic bounds to compute expected values. For example, please refer to (Marlin et al., 2011). Our experiments adopt this technique and replicate the setup outlined in (Khan & Lin, 2017).

Table 2: Dataset Statistics and Model Hyperparameters

| Dataset | $\beta$ | $N$ | $d$ | $N_{\text{train}}$ | Step Sizes | | |
|---|---|---|---|---|---|---|---|
| | | | | | VN | SQ-VN | BW-GD |
| Australian-scale | $10^{-5}$ | 690 | 14 | 552 | $5 \times 10^{-3}$ | $5 \times 10^{-3}$ | $4.4 \times 10^{-3}$ |
| Diabetes-scale | $10^{-2}$ | 768 | 8 | 614 | $5 \times 10^{-3}$ | $5 \times 10^{-3}$ | $9 \times 10^{-4}$ |
| breast-cancer | $10^{-1}$ | 683 | 10 | 546 | $9 \times 10^{-3}$ | $6 \times 10^{-3}$ | $6.3 \times 10^{-3}$ |
| Mushrooms | $10^{-2}$ | 8124 | 112 | 6499 | $2.5 \times 10^{-4}$ | $2.5 \times 10^{-4}$ | $8.5 \times 10^{-5}$ |
| Phishing | $10^{-2}$ | 11055 | 68 | 8844 | $4 \times 10^{-4}$ | $4 \times 10^{-4}$ | $8 \times 10^{-5}$ |
| MNIST | $10^{-1}$ | 70000 | 784 | 60000 | $5 \times 10^{-6}$ | $5 \times 10^{-6}$ | $10^{-6}$ |
| Covtype-scale | $2 \times 10^{-2}$ | 581012 | 54 | 500000 | $10^{-5}$ | $10^{-5}$ | $10^{-6}$ |
| Leukemia | $2 \times 10^{-1}$ | 38 | 7129 | 34 | $5 \times 10^{-6}$ | $5 \times 10^{-6}$ | $1.5 \times 10^{-6}$ |

### F.1 Runtime comparison

It is worth noting that each algorithm differs in computational complexity *not* because of the training data size—which is the same across all—but due to its specific update operations after computing the expected gradient **g** and Hessian **H**. Since these quantities cost the same for each method, we focus on the subsequent matrix/vector steps.

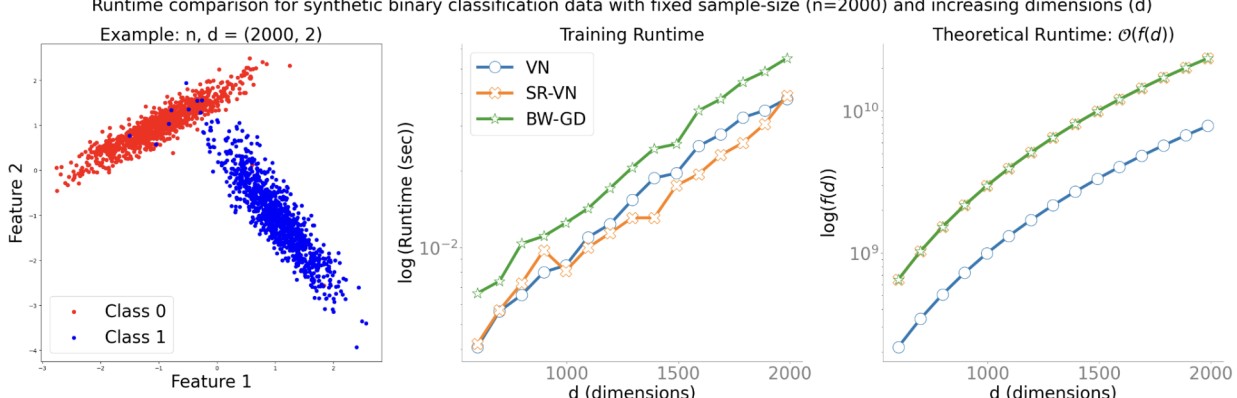

Figure 5: We compared the run times of all three algorithms across increasing dimensions while keeping the number of data points ($n$) fixed. The run times were averaged over 1000 iterations per algorithm. The theoretical run times, derived in our rebuttal answer, are functions of $d$ and are plotted to reflect the floating point operations. The results indicate that the run times can vary significantly among the algorithms, depending on the specific data and dimensionality.

1. **Complexity for VN:**
    - **S** update: scalar multiplication + matrix addition $\rightarrow \mathcal{O}(2d^2)$.
    - **m** update: matrix inversion $\rightarrow \mathcal{O}(d^3)$, plus matrix-vector product $\rightarrow \mathcal{O}(d^2)$, plus matrix addition $\rightarrow \mathcal{O}(d^2)$, for a total $\mathcal{O}(2d^2 + d^3)$.
    - **Overall cost:** $\mathcal{O}(d^3 + 4d^2)$.

2. **Complexity for SR-VN:**
    - **C** update: includes $\mathbf{C}^\top \mathbf{H} \mathbf{C}$ ($\mathcal{O}(d^3)$) plus some matrix additions/subtractions ($\mathcal{O}(d^2)$) and multiplications ($\mathcal{O}(d^3)$).
    - **m** update: $\mathbf{C}\mathbf{C}^\top$ multiplication ($\mathcal{O}(d^3)$) plus matrix-vector products ($\mathcal{O}(d^2)$).
    - **Overall cost:** $\mathcal{O}(3\,d^3 + \frac{9}{2}\,d^2 + \frac{d}{2})$.

3. **Complexity for BW-GD:**
    - **m** update: scalar multiplication + matrix subtraction $\rightarrow \mathcal{O}(3d^2)$.
    - **M** update: two matrix additions + scalar operations $\rightarrow \mathcal{O}(2d^2)$.
    - **V** update: matrix inversion $\rightarrow \mathcal{O}(d^3)$ plus two matrix multiplications $\rightarrow \mathcal{O}(2d^3)$, totaling $\mathcal{O}(3d^3)$.
    - **Overall cost:** $\mathcal{O}(3d^3 + 5d^2)$.

These matrix/vector operations can dominate the wall-clock time for each algorithm. To illustrate how runtime scales in practice, we compare VN, SR-VN, and BW-GD in Figure 5 as the dimension grows, highlighting their differing execution costs.

### F.2 Algorithmic Details and Additional Results

In Table 3, we provide descriptive statistics for different metrics achieved by each algorithm when it is run for every dataset considered.

Table 3: Performance Comparison on different Datasets

| Dataset | Algorithm | Train ELBO | Test NLL | Test Accuracy | Time (sec) |
|---|---|---|---|---|---|
| Australian-scale | VN | 196.82 | 54.61 | 0.877 | 0.53 |
| | BW-GD | 196.82 | 54.61 | 0.877 | 0.53 |
| | SQ-VN | 196.82 | 54.61 | 0.877 | 0.53 |
| Diabetes-scale | VN | 301.18 | 79.72 | 0.74 | 0.7 |
| | BW-GD | 301.18 | 79.72 | 0.74 | 0.49 |
| | SQ-VN | 301.18 | 79.72 | 0.74 | 0.53 |
| Breast-cancer | VN | 52.86 | 13.62 | 0.956 | 0.61 |
| | BW-GD | 52.86 | 13.62 | 0.956 | 0.61 |
| | SQ-VN | 52.86 | 13.62 | 0.956 | 0.61 |
| mushrooms | VN | 57.51 | 6.69 | 1.0 | 1.04 |
| | BW-GD | 78.41 | 7.02 | 0.999 | 1.04 |
| | SQ-VN | 53.56 | 6.43 | 0.999 | 1.05 |
| Leukeumia | VN | 339.41 | 727.96 | 0.706 | 0.36 |
| | BW-GD | 564.65 | 796.76 | 0.676 | 0.38 |
| | SQ-VN | 339.38 | 727.99 | 0.706 | 0.37 |
| Phishing | VN | 1267.77 | 334.19 | 0.938 | 0.99 |
| | BW-GD | 1377.645 | 349.10 | 0.933 | 0.97 |
| | SQ-VN | 1268.06 | 333.37 | 0.939 | 0.98 |
| MNIST | VN | 6317.52 | 1017.64 | 0.989 | 0.22 |
| | BW-GD | 6367.23 | 1102.34 | 0.988 | 0.23 |
| | SQ-VN | 6301.87 | 1012.37 | 0.989 | 0.2 |
| Covtype-scale | VN | 27066.64 | 10741.37 | 0.745 | 0.95 |
| | BW-GD | 28036.23 | 11259.20 | 0.714 | 0.98 |
| | SQ-VN | 27065.02 | 10740.73 | 0.745 | 0.91 |

## F.3 Iteration Plots with Accuracy

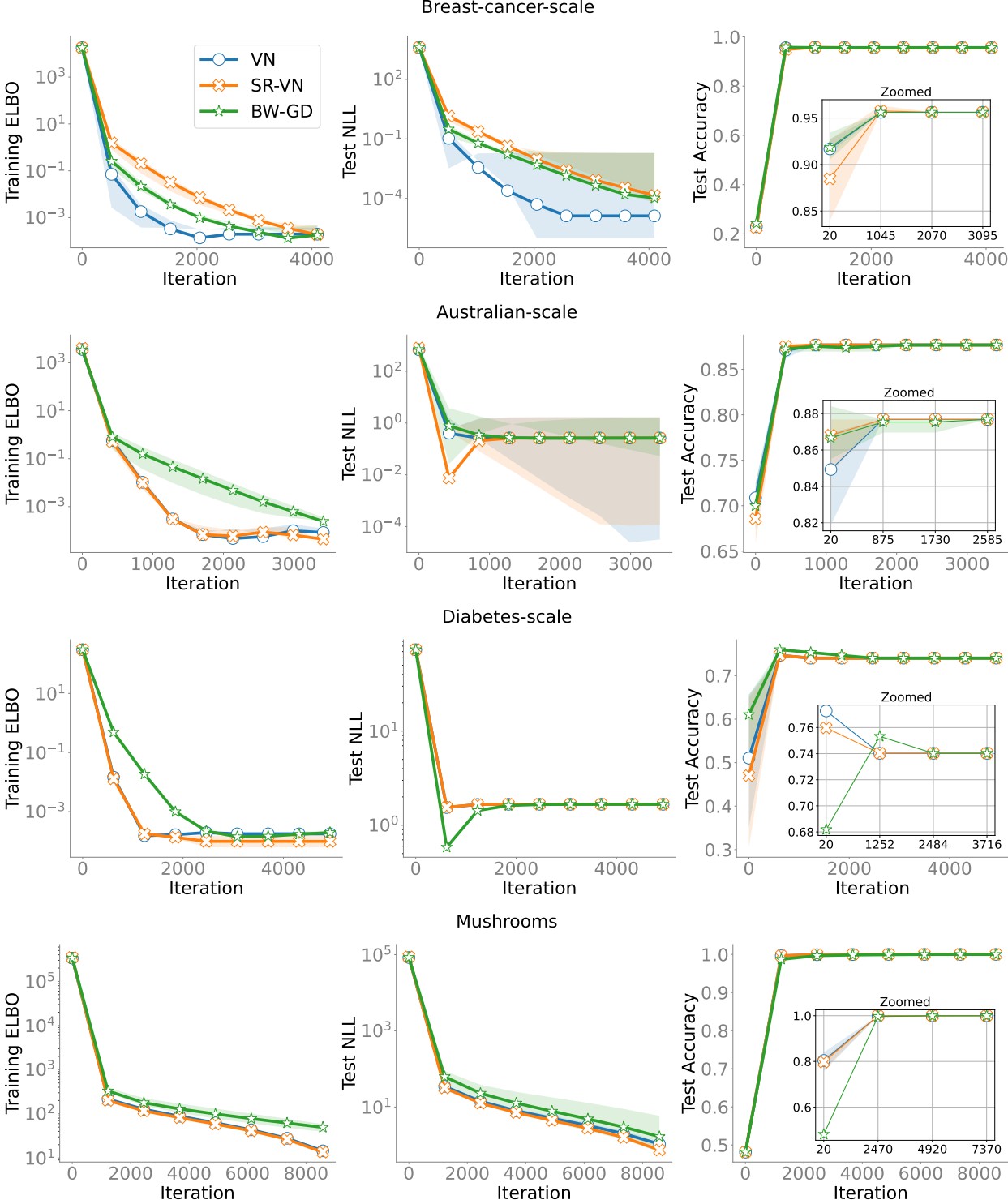

Figure 6: Plots with respect to the number of iterations for small-scale datasets.

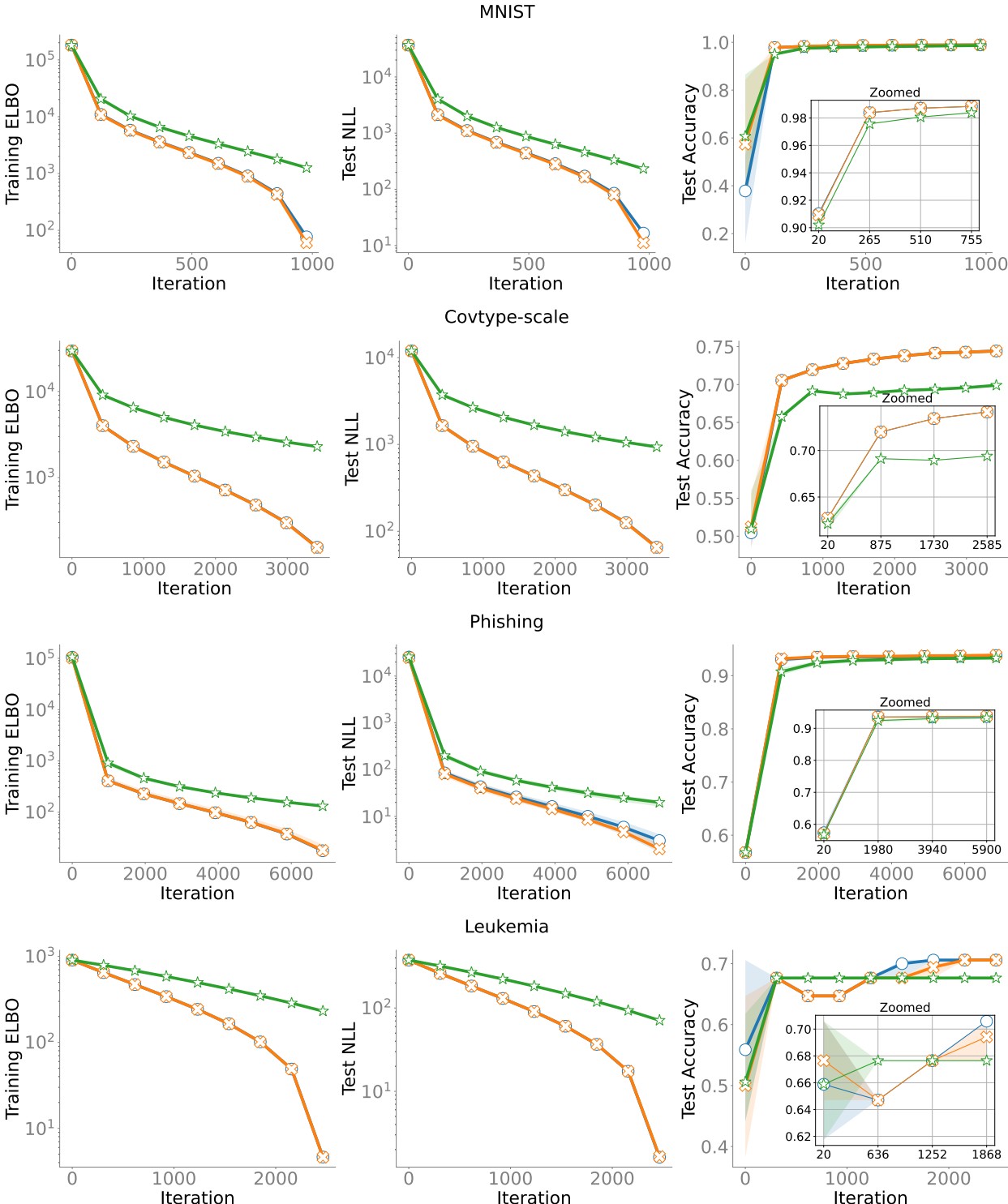

Figure 7: Plots with respect to the number of iterations for large-scale datasets.

## F.4 Time Plots with Accuracy

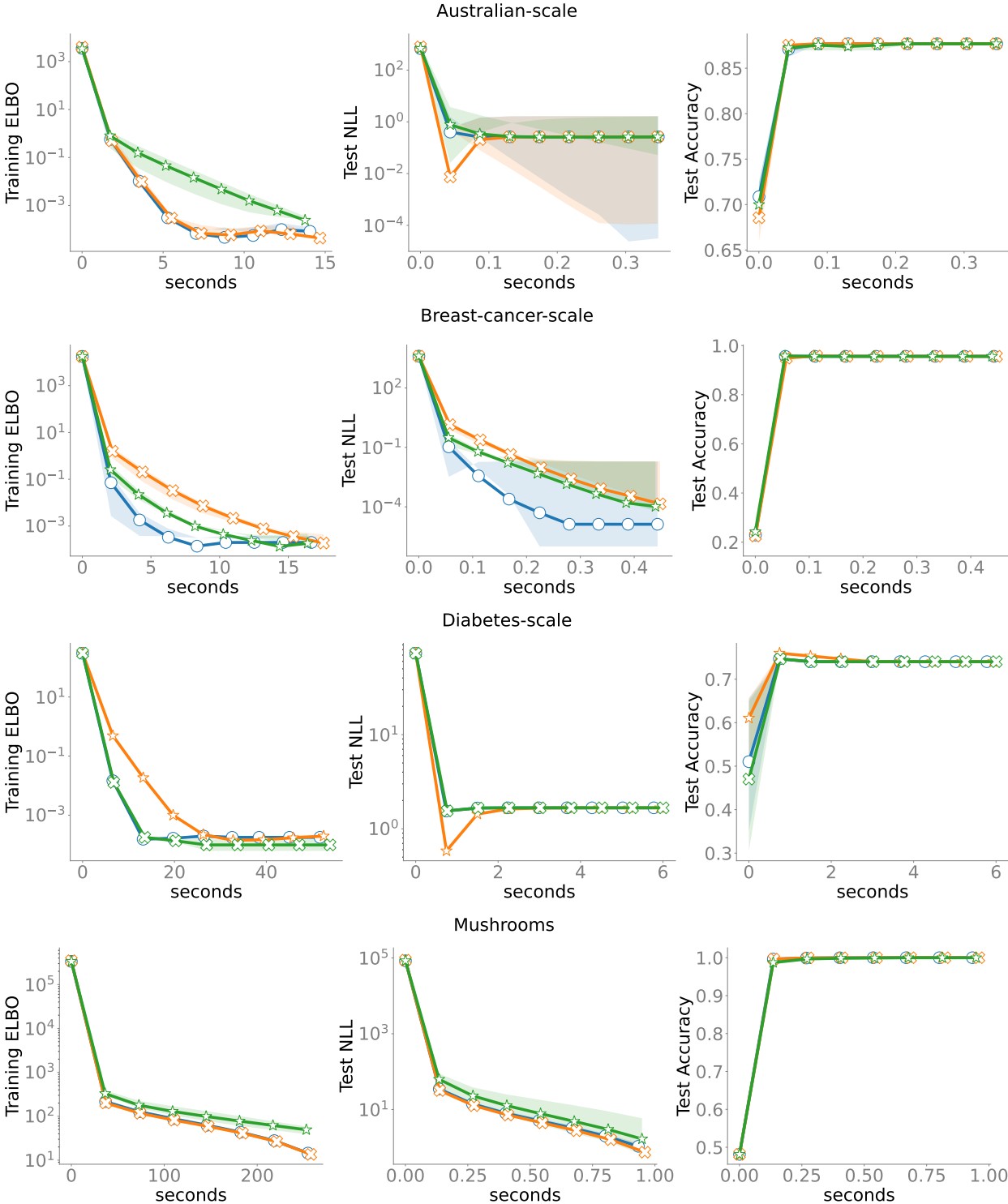

Figure 8: Plots with respect to time for small-scale datasets.

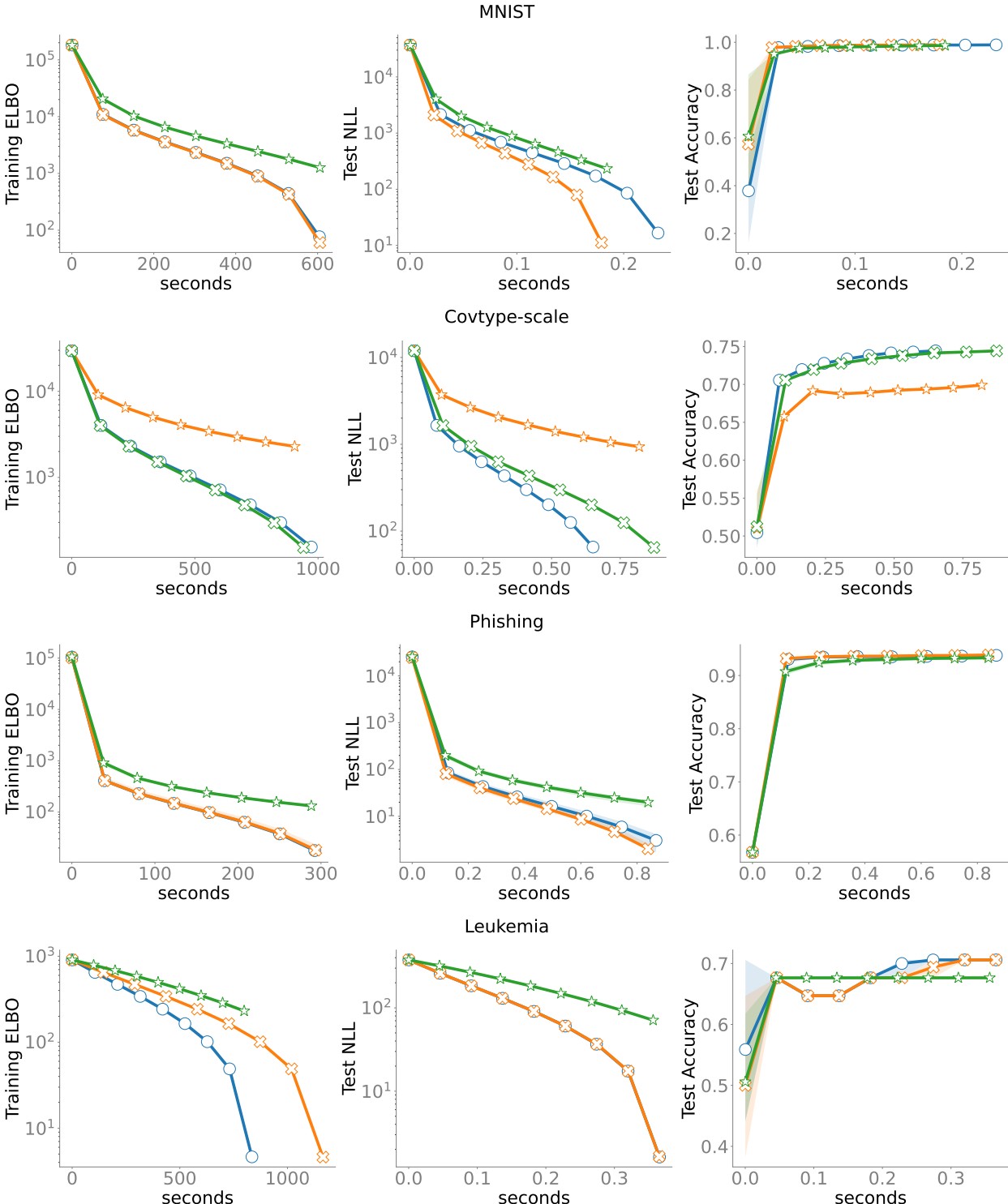

Figure 9: Plots with respect to time for large-scale datasets.

# G   Comparing Geometries: SR-VN vs. Alternatives

In this section, we discuss two complementary comparisons for SR-VN: (1) a theoretical analysis of its convergence rate relative to BW-GD, and (2) an empirical evaluation against an enhanced method tailored for the BW geometry.

**Theoretical rates comparison, SR-VN vs BW-GD:**   To estimate the convergence rate for BW-GD, we refer to (Lambert et al., 2022b, Theorem 4), which—under the assumption of strong log-concavity—provides an exponential rate in the Wasserstein distance once we ignore the variance term because of its deterministic nature. Consequently, in terms of asymptotic guarantees, BW-GD and SR-VN both exhibit exponential convergence rates. However, the specific details—namely, **which geometry** (Fisher–Rao vs. Wasserstein) is deployed, and **which measure** (KL in parameter space vs. Wasserstein distance in probability space) is analyzed—separate these methods apart. As a result, establishing a theoretical comparison is challenging, leading us to focus on experimental evidence in the next section.

**Additional experimental comparisons.**   Although BW-GD is a natural baseline for SR-VN in the BW geometry, more recent studies have proposed improved and more stable stochastic BW methods (e.g., Diao et al. (2023); Liu et al. (2024)). We focus here on Liu et al. (2024), which uses SVGD and has demonstrated better performance than Diao et al. (2023). Specifically, we implemented a **deterministic** version of (Liu et al., 2024, Algorithm 1)—i.e., replacing sample estimates with the full expectation—and refer to this method as **SVGD-density**. It is worth mentioning that Liu et al. (2024) demonstrated improvements in the stochastic setting compared to BW-GD. However, since we focus on the deterministic implementation here, it remains unclear whether the same advantages apply—this is precisely what we aim to address next.

We evaluated SVGD-density (using various step sizes) on the same 2D Bayesian linear regression problem from Figure 1, as shown in Figure 10 (upper). Additionally, we tested it on the Bayesian Logistic Regression scenario in Figure 3, as shown in Figure 10 (lower). In both plots, SVGD-density shows the slowest convergence among the compared algorithms. Although it can approach the performance of BW-GD, given a finely tuned step size e.g. in Figure 10 (upper), its performance is generally worse compared to SR-VN or VN.

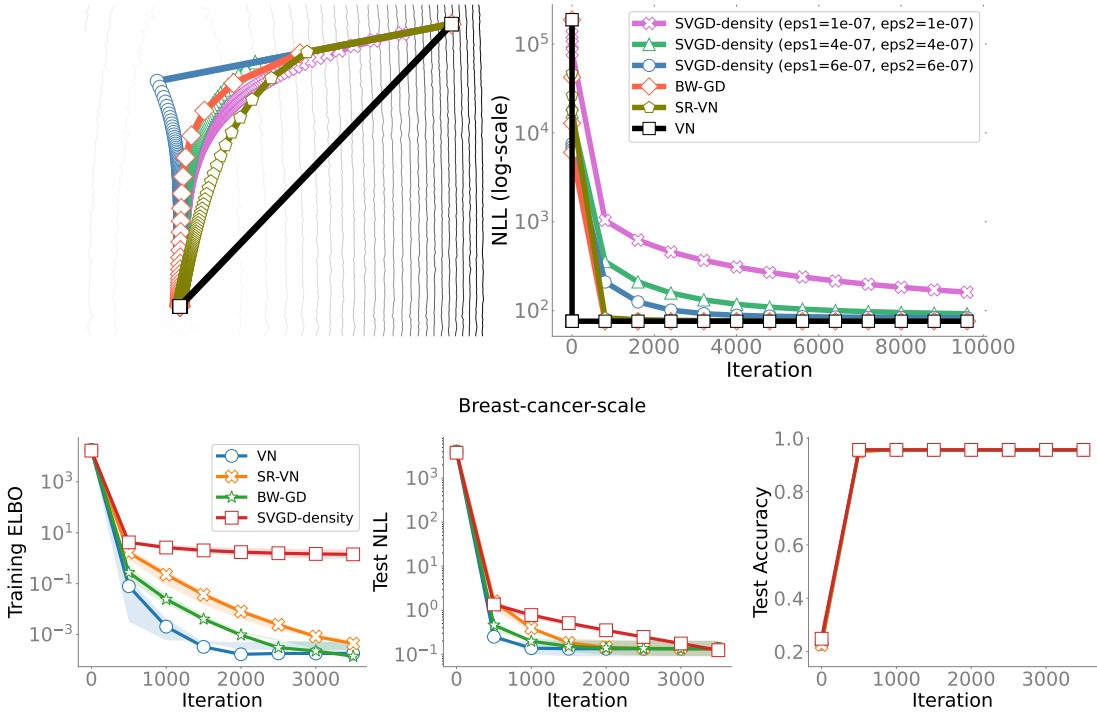

Figure 10: Comparison of SVGD-density: Bayesian linear regression (top) and Bayesian logistic regression (bottom).

## H  Non-convex Model

In this section, we evaluate all algorithms on the same Bayesian logistic regression loss in Equation (15) with the addition of a non-convex regularizer $\sum_{i=1}^{d} m_i^2/(1 + m_i^2)$.

The results in Figure 11 indicate that all compared methods still converge despite the non-convexity introduced in the mean parameter, suggesting promising directions for further investigation into non-convex scenarios.

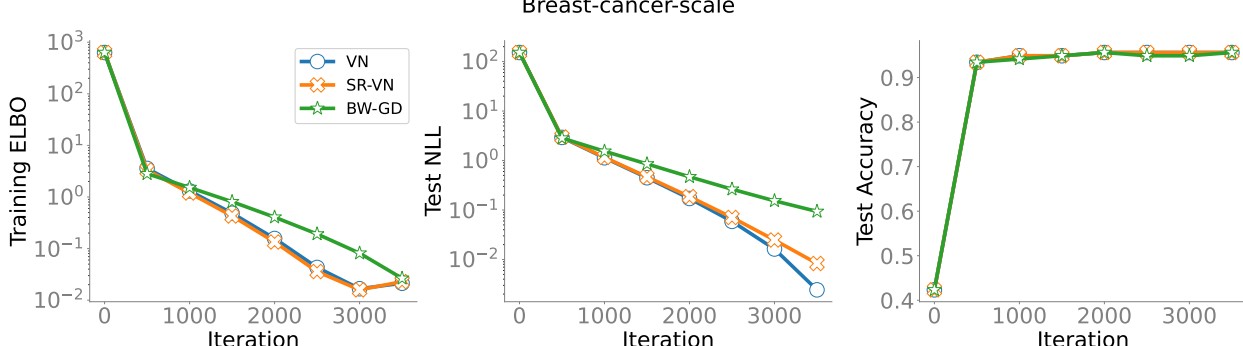

Figure 11: To evaluate performance under non-convex conditions, we augment the Bayesian logistic regression problem in Equation (15) with a non-convex regularization term: $\sum_{i=1}^{d} m_i^2/(1 + m_i^2)$.

