# OpenReview forum: "Optimization Guarantees for Square-Root Natural-Gradient Variational Inference"
_TMLR — Accepted by TMLR_

### Review · Reviewer_442L · 2025-01-04

**Summary Of Contributions:**

Variational Inference is a common technique to approximate posterior distribution. However, it is not theoretically well understood. This paper focuses on Variational Inference with the approximation belonging to the Gaussian family, and studies a Natural-Gradient algorithm to minimize the Kullback-Leibler divergence over this family. It is shown that while the minimization problem is not convex in the mean and covariance even when the target distribution is log-concave, it is with respect to the mean and the square root of the covariance. Leveraging this observation, the authors show that the Natural Gradient Flow converges exponentially fast, showing that the Kullback-Leibler divergence satisfies a Riemannian PL inequality. Moreover, they propose a discretization of the Natural Gradient Flow on the mean and the square root of the covariance. Then, they show that this algorithm converges towards the minimum under reasonable assumptions (strong convexity and smoothness of the loss). Finally, they apply it to a Bayesian Logistic Regression problem, and show that the proposed algorithm has the same performances as the Natural Gradient Descent Variational Inference algorithm, and better performances than the Bures-Wasserstein Gradient Descent.

**Audience:**

Yes

**Broader Impact Concerns:**

None.

**Claims And Evidence:**

Yes

**Requested Changes:**

These are not change which would be crucial for securing my recommendation for acceptance, but it might strengthen the work.

- Since you compare with BW-GD, it would be a nice addition to mention the convergence rates known for this method, and compare it with the convergence rate derived in Theorem 1 and 2.
- On the applications, it could be nice to have some comparison on a simple Gaussian target (in the same spirit as Figure 1, but e.g. also plotting the convergence), additionally to the Bayesian Logistic Regression.
- In relation with the weakness I mentioned, it could be nice to compare with or at least mention [1], which proposes a scheme to solve this problem which is more stable than BW-GD. I do not expect it to have better results though. Another work, which could be pertinent to mention and/or compare with, is [2], which uses SVGD. They also applied their scheme to Bayesian Logistic Regression, and demonstrated better results than [1].
- Less related, but note that there are some works trying to improve the convergence of Wasserstein gradient descent by changing the geometry of the space, see e.g. [3].


[1] Diao, M. Z., Balasubramanian, K., Chewi, S., & Salim, A. (2023). Forward-backward Gaussian variational inference via JKO in the Bures-Wasserstein space. In International Conference on Machine Learning (pp. 7960-7991). PMLR.

[2] Liu, T., Ghosal, P., Balasubramanian, K., & Pillai, N. (2024). Towards understanding the dynamics of gaussian-stein variational gradient descent. Advances in Neural Information Processing Systems, 36.

[3] Cheng, Z., Zhang, S., Yu, L., & Zhang, C. (2024). Particle-based variational inference with generalized wasserstein gradient flow. Advances in Neural Information Processing Systems, 36.

**Strengths And Weaknesses:**

**Strengths:**

The paper studies theoretically the convergence of Natural Gradient Descent Variational Inference algorithm. It provides new insights on the convexity of the problem, shows the convergence of the Natural Gradient Flow, and provides a new discretization of the flow with a parametrization using the mean and the square root of the covariance, which is shown to converge. This new findings are very interesting, and provide a nice justification of the convergence of Variational Inference algorithm with Natural Gradient Descent. Moreover, the paper is very clear and pleasant to read.


**Weaknesses:**

A little weakness might be the choice of baselines in the Bayesian Logistic Regression problem. While it makes senses to compare the algorithm with the Bures-Wasserstein gradient descent, there are some more recent works which focused also on this problem. Also, I would be curious to see a comparison of the theoretical guarantees for the Bures-Wasserstein gradient descent with those derived in this work.

---

> ### Author Response · Authors · 2025-01-20
> **Response to your requested changes**
>
> Thank you for your thoughtful comments regarding additional baselines and theoretical comparisons. We appreciate the suggestions and have **integrated** the following enhancements into the **new version** of our submission, in particular, please see Appendix G:
>
> 1. **Comparison with Bures–Wasserstein Gradient Descent (BW-GD) Theoretical Rates**
>    We agree that a more direct statement on BW-GD’s known rates—alongside our Theorems 1 and 2—clarifies the discussion. In this revised version, we include a brief comparison of the convergence properties of BW-GD and our approach, highlighting both shared assumptions and distinct iteration complexities:
>
>    - **Continuous Case**
>      - **Exponential Convergence**: Both analyses show that strong log-convexity in the chosen geometry yields exponential convergence in time to the optimum in the variational family.
>      - **Finite-vs.-Infinite Dimensional**: SR‑VN evolves effectively as an ODE in a *finite-dimensional* parameter space $(\mathbf{m}, \mathbf{C})$, whereas the BW flow is formulated as an ODE in the *infinite-dimensional* space of probability measures (specialized to Gaussians).
>      - **Which Metric or Gap**: SR‑VN measures progress via the KL gap within the parametric family; BW flows track $\mathrm{KL}$ or the Wasserstein distance ($W_2$) in the probability distribution space. Both exhibit $\exp(-~\text{const}\cdot t)$ rates under parallel assumptions of strong log-concavity.
>
>      Thus, while the *continuous-time viewpoint* in each method ensures an exponential rate (in terms of time), the specifics$—$**which geometry** is employed (Fisher-Rao vs. Wasserstein) and **which measure** of convergence is proven (KL in parameter space vs. $W_2$ in probability space)$-$differentiate the SR‑VN flow from the BW flow.
>
>    - **Discrete Case**
>      A direct theoretical comparison between SR‑VN and BW‑**S**GD is infeasible because they operate in fundamentally different geometric frameworks, with SR‑VN being deterministic and BW‑SGD being stochastic. We therefore turned to an empirical comparison as detailed below:
>      - To facilitate a meaningful comparison in our experiments, we employed a *deterministic* variant of BW‑SGD, **BW‑GD**, which uses the full expected gradient and Hessian of the NLL rather than single-sample estimates (as in Algorithm 1 of [1]). Consequently, if we were to estimate the convergence rate for BW‑GD according to Theorem 4 in [1], we could simply ignore the variance term to obtain an exponential rate for its convergence in Wasserstein distance. This rate is analogous to the exponential convergence observed for SR‑VN (Theorem 2 in our paper).
>      - Beyond this point, one has to rely on experimentation to decide which geometric viewpoint is more suitable in practice. Our experimental results clearly demonstrate that SR-VN exhibits better performance on the tested datasets (see Figures 2 and 3).
>
> 2. **Comparison with SVGD**
>    - As requested, we now compare **SR‑VN**, **VN**, and **SVGD-density** ([2], Algorithm 1) in Figure 10 of our manuscript.
>    - Although our primary focus was not on methods in the Wasserstein geometry, we have included a short discussion of these approaches in our revised manuscript.
>
> Thank you again for your constructive comments and for pointing us to these recent works (which are now cited in our revision). We believe these additions significantly strengthen our new submission.
>
> ---
>
> **References**
>
> [1] Lambert, M., Chewi, S., Bach, F., Bonnabel, S., & Rigollet, P. (2022). Variational inference via Wasserstein gradient flows. *Advances in Neural Information Processing Systems*, 35, 14434–14447.
>
> [2] Liu, T., Ghosal, P., Balasubramanian, K., & Pillai, N. (2024). Towards understanding the dynamics of gaussian-stein variational gradient descent. *Advances in Neural Information Processing Systems*, 36.

---

### Review · Reviewer_Cj54 · 2025-01-22

**Summary Of Contributions:**

The paper provides a theoretical analysis of the convergence of Natural Gradient methods for Variational Inference, in the context of a Gaussian square root parametrization, i.e. Variational Inference with a Gaussian distribution and using a Cholesky decomposition of the covariance matrix. This setting provides significant advantages, e.g. maintaining concavity of the objective which allows to use tools from convex optimization for the analysis.
After introducing some background on Variational Inference and Natural Gradient methods (Natural Gradient Descent and its continuous form Natural Gradient Flow), the paper presents its main algorithm, SR-VN (Square Root Variational Newton), using this Cholesky decomposition to derive a new update for a Newton method. It also  provides theoretical guarantees of its exponential convergence in the case of a strongly concave log-likelihood. It finally presents some empirical results on Bayesian Logistic Regression problems.

**Audience:**

Yes

**Claims And Evidence:**

Yes

**Requested Changes:**

See weaknesses section.

**Strengths And Weaknesses:**

Strenghts:
- Background on Variational Inference and NG methods is well explained
- In general, explanations of the paper are clear and intuitive
- Square root Gaussian parametrization is an elegant solution to address the lack of concavity issue
- Theoretical analysis is rigorous and provides interesting bounds on SR-VN
- Paper is clearly organized - e.g. background, assumptions, main results, experiments - which makes the whole proof and reasoning easy to follow

Weaknesses:
- No proper literature review - section 2 cites briefly some previous work done in the general area but I would have wanted a much clearer review of what has been done in this specific topic
- Fairly limited discussion section - I would have expected much more discussion on the experiment result as well as on the strenghts of the assumptions made in section 4

---

> ### Author Response · Authors · 2025-02-10
> **Added changes based on your suggestions**
>
> We appreciate the reviewer’s comments and value the recommendations provided to enhance our work. In response, we have provided a revised manuscript, where we have added Section 7 (Related Works) and expanded Section 8 (Discussion and Conclusion).
>
>  - Section 7 provides a more comprehensive overview of the existing literature on NGVI, which helps us to contextualize our work by discussing different NGVI approaches, highlighting the ongoing research in convergence analysis for these methods, and emphasizing the crucial role of parameterization choices.
>
>  - Section 8 now offers a more detailed analysis of our experimental findings, highlighting the impact of parameterization and optimization geometry on the performance of NG flow and NGD. We expand on how the square-root parameterization, validated empirically, contributes to stable updates and comparable performance. Finally, we openly discuss the limitations of our current study and outline key directions for future research, including extensions to stochastic NGVI and reassessment of our assumptions in more complex settings.
>
> We believe these additions provide a more transparent and nuanced understanding of our work in relation to the broader field.

---

### Review · Reviewer_TQto · 2025-02-23

**Summary Of Contributions:**

This paper studies NGD for gaussian variational inference. It uses the previously establisehd fact that for a concave loglikelihood, the variational objective is convex wrt square root parameterization of the covariance, but not wrt to the covariance itself or the precision. The main contributions are to apply convex optimization tools to the square-root parameterization to prove convergence for NG flow and for NGD discretized wrt the Cholesky factor.

**Audience:**

Yes

**Claims And Evidence:**

Yes

**Requested Changes:**

I think the paper would benefit from an experiment on a non-convex model. There’s value in papers that prove theoretical guarantees in restricted settings and test empirically whether performance tends to extend beyond those assumptions. On the other hand a paper that only addresses convex models is rather limited.

Parameterization does not matter for NG flow. The introduction is careful on this point, only stating the algorithm depends on the parameterization in the discretized case, and the end of sec 3.3 explains the square root parameterization is used in the flow case only as a proof technique since it allows application of convex analysis. Still it could improve clarity to be more explicit about this in the introduction.

After eq 1, the description of $\gamma$ seems to be reversed. Setting $\gamma$ too high should cause the posterior to collapse to the prior. Setting $\gamma$ too low should produce no regularization and hence excess sampling variance.

The remark before eq 5 that $\boldsymbol{\tau}$ is the natural parameter is unnecessary. The equation holds regardless. On the other hand the choice of natural parameters is essential for eq 7.

Sec 3.2: It may help to highlight that the Fisher of $q_{\boldsymbol{\tau}}(\boldsymbol{\theta})$ used here is different from the Fisher of $p(\boldsymbol{y}|\boldsymbol{\theta})$ used in non-variational NGD (especially since terms related to the latter appear in eq 6).

The statement that eq 8 is a continuous-time version of eq 5 holds only when $\gamma=1$.

The statement after eq 9 could be made more precise: “Upon discretization with respect to the natural parameters, this results in the VN update with $\gamma=1$.”

Sec 3.5: It may help to point out that, in addition to its connection to Newton, eq 7 with $\gamma=\rho=1$ is the exact Bayes update in the conjugate case, while discretizations based on $\boldsymbol{V}$ or $\boldsymbol{C}$ are only approximations.

Eq 15: missing $-$ and $)$

After eq 15: The Hessian is $\boldsymbol{X}^\top \boldsymbol{D} \boldsymbol{X}$ with $\boldsymbol{D}\in\mathbb{R}^{n\times n}$.

same paragraph: the loss is $\beta$-strongly convex, not $\delta$-strongly

Assumption 3: The citation should be to Domke 2020 and the premise should be that $\bar{\ell}$ is $M$-Lipschitz-smooth

Assumption 4 for logistic regression: It’s not obvious to me that the bounds on $\boldsymbol{H}$ imply bounds on $\boldsymbol{C}$. The reasoning "This validates the assumption of a bounded square root” is too casual since $\boldsymbol{C}$ is not the square root of $\boldsymbol{H}$ but instead is the result of the update in Eq 14.

Sec 6.1 NLL definition: sum should run to $n_{\rm test}$ and missing $-$ sign

Covtype-scale: extra commas in $n$ and $n_{\rm train}$

"Our findings indicate that the success of these methods is heavily influenced by the choice of parameterization”: I don’t see this in the results. The parameterization matters for proof techniques but does not necessarily matter for performance. The experiments mainly show the natural and square-root parameterizations perform comparably, except in the conjugate case where natural parameterization gives the exact Bayes update (fig 1c).

**Strengths And Weaknesses:**

Strengths:
* First rigorous convergence guarantee for discrete-time variational NGD

Weaknesses:
* The main new update, eq 14, is already published by Tan (2025)
* The assumption of convex loglikelihood limits applicability, excluding most ML settings of interest
* The story is incomplete: SR-VN makes the problem convex but VN still performs about as well. Why?

---

> ### Author Response · Authors · 2025-03-09
>
> We thank the reviewer for thoroughly reviewing our manuscript and providing valuable feedback. Below, we address your comments on the “weaknesses” individually.
>
> > The main new update, eq 14, is already published by Tan (2025)
>
>  -  Indeed, the update (Eq. 14) was previously published by Tan (2025). However, our contribution is novel due to the unique analysis and interpretation we provide, which sheds new light on its behavior and performance under different conditions, previously not discussed by Tan.
>
> > The assumption of convex loglikelihood limits applicability, excluding most ML settings of interest
>
>  - While it is true that our analysis relies on convex log-likelihood assumptions, we would like to point out that convex optimization still remains widely relevant in machine learning, underpinning many standard algorithms such as logistic regression, support vector machines, and certain neural network architectures. Investigating these convex scenarios serves as a valuable first step toward understanding the deeper mechanisms of the algorithms studied.
>
> > The story is incomplete: SR-VN makes the problem convex but VN still performs about as well. Why?
>
>  - Our experiments indeed show that VN maintains comparable or better (in some cases) performance to SR-VN despite the convexity introduced by SR-VN. One possibility is that VN inherently employs a more optimal strategy under certain conditions; for instance, in the two-dimensional case, VN converges in a single step. Additionally, the effectiveness of VN could be attributed to its relationship with Newton-type algorithms, which are known for their rapid convergence—this connection has been discussed in [1].  Nonetheless, understanding precisely why VN outperforms SR-VN in broader scenarios remains an interesting and open question for future research.
>
> [1] Khan, M. E., & Rue, H. (2023). The Bayesian learning rule. Journal of Machine Learning Research, 24(281), 1-46.
>
>
> > I think the paper would benefit from an experiment on a non-convex model. There’s value in papers that prove theoretical guarantees in restricted settings and test empirically whether performance tends to extend beyond those assumptions. On the other hand, a paper that only addresses convex models is rather limited.
>
>  - We appreciate this suggestion and agree that extending the empirical evaluation to non-convex scenarios would strengthen our findings. We therefore followed your suggestion and incorporated a non-convex regularizer $\epsilon \sum_{i=1}^d m_i^2/(1 + m_i^2)$ (used in several prior works, we chose $\epsilon=1$ here) to obtain a non-convex function and evaluated its performance on the breast-cancer-scale dataset (please see the supplementary material). Our results demonstrate promising performance and hope this addresses the reviewer's comment.
>
> ---
> Finally, we will fix all the minor changes requested by the reviewer in the final version of our manuscript. Once again, we greatly appreciate the reviewer’s suggestions, which have helped us strengthen our work.

---

### Public Comment · ~Wu_Lin1 · 2025-04-08

Dear authors,

I would like to bring to your attention that your Algorithm 1 is also considered in my ICML 2023 and ICML 2024 papers. For example, please see Eq 6 of [1].
Eq 6 recovers your Algorithm 1 when using a first-order truncation of the matrix exponential $Exp(N) \approx  I+N$. I use this truncation in my ICML 2023 and ICML 2024 papers.

* [1] Lin et al, Structured inverse-free natural gradient: Memory-efficient & numerically-stable kfac, ICML 2024, https://proceedings.mlr.press/v235/lin24f.html
* [2] Lin et al, Simplifying momentum-based positive-definite submanifold optimization with applications to deep learning, ICML 2023, https://proceedings.mlr.press/v202/lin23c.html

---

> ### Author Response · Authors · 2025-04-11
>
> Thank you for bringing your works to our attention! This is very interesting to us. The resultant update aligns with independently proposed updates by Tan (2025). We have gladly cited your papers in the revised version.
>
> [1] Tan, L. S. (2025). Analytic natural gradient updates for Cholesky factor in Gaussian variational approximation. Journal of the Royal Statistical Society Series B: Statistical Methodology.

---

### Public Comment · ~Kyurae_Kim1 · 2025-06-30

Dear Authors,

Congratulations for acceptance! I was taking a quick look but it seems that some other reference was intended in place of Ko *at al.* (2024) in page 11? Perhaps the batch-and-match paper given the context?

---

> ### Author Response · Authors · 2025-07-15
>
> Thank you for your comment. You are indeed correct regarding the citation error—we appreciate it for pointing it out. It has been fixed in our arXiv version, available here: https://arxiv.org/abs/2507.07853

---

### Decision · Action_Editor_EFRb · 2025-04-28

**Recommendation:** Accept with minor revision

**Comment:**

Reviewers unanimously recommend acceptance. Please see above for motivation of request for minor revision.

**Audience:**

Yes, reviewers are in agreement that this submission is of sufficient interest to the community.

**Claims And Evidence:**

The main theoretical contribution in this submission are guarantees of convergence to for natural gradient (NG) flow and for natural gradient descent that is discretized w.r.t. the Cholesky factor. Reviewers are in agreement that claims with respect to these contributions are sufficiently supported.

A point of contention that the authors appear not to have addressed is the claim (found in the discussion) that:

> Our findings indicate that the success of these methods is heavily influenced by the choice
of parameterization and the intrinsic geometry of the optimization landscape

As noted by reviewer TQto, this claim does not appear to be resulted by experimental results show very similar performance for variational Newton (VN) and the Square-Root VN (SR-VN) for which the authors obtain theoretical results. This seems to also be acknowledged by the authors in the discussion. In this context, a minor revision to reformulate this language seems warranted.